# Descending locus coeruleus noradrenergic signaling to spinal astrocyte subset is required for stress-induced mechanical pain hypersensitivity

**Riku Kawanabe-Kobayashi[1†], Sawako Uchiyama[1†], Kohei Yoshihara[1†], Keisuke Koga[2], Daiki Kojima[1], Thomas J McHugh[3], Izuho Hatada[4,5], Ko Matsui[6], Kenji F Tanaka[7], Makoto Tsuda[1,8]\***

[1]Department of Molecular and System Pharmacology, Graduate School of Pharmaceutical Sciences, Kyushu University, Fukuoka, Japan; [2]Department of Neurophysiology, Hyogo Medical University, Nishinomiya, Japan; [3]Laboratory for Circuit and Behavioral Physiology, RIKEN Center for Brain Science, Wako, Japan; [4]Laboratory of Genome Science, Biosignal Genome Resource Center, Institute for Molecular and Cellular Regulation, Gunma University, Maebashi, Japan; [5]Viral Vector Core, Gunma University Initiative for Advanced Research (GIAR), Maebashi, Japan; [6]Super-network Brain Physiology, Graduate School of Life Sciences, Tohoku University, Sendai, Japan; [7]Division of Brain Sciences, Institute for Advanced Medical Research, Keio University School of Medicine, Tokyo, Japan; [8]Kyushu University Institute for Advanced Study, Fukuoka, Japan

*For correspondence:
tsuda@phar.kyushu-u.ac.jp

†These authors contributed equally to this work

## eLife Assessment

This **important** study identifies a novel role for Hes5+ astrocytes in modulating the activity of descending pain-inhibitory noradrenergic neurons from the locus coeruleus during stress-induced pain facilitation. The role of glia in modulating neurological circuits including pain is poorly understood, and in that light, the role of Hes5+ astrocytes in this circuit is a key finding with broader potential impacts. This work is supported by **convincing** evidence, albeit somewhat limited by the indirect nature of the evidence linking adenosine to nearby neuronal modulation, and possible questions on the population specificity of the transgenic approach.

**Abstract** It is known that stress powerfully alters pain, but its underlying mechanisms remain elusive. Here, we identified a circuit, locus coeruleus descending noradrenergic neurons projecting to the spinal dorsal horn (LC$^{\rightarrow SDH}$-NA neurons), that is activated by acute exposure to restraint stress and is required for stress-induced mechanical pain hypersensitivity in mice. Interestingly, the primary target of spinal NA released from descending LC$^{\rightarrow SDH}$-NAergic terminals causing the stress-induced pain hypersensitivity was $\alpha_{1A}$-adrenaline receptors ($\alpha_{1A}$Rs) in *Hes5*-positive (*Hes5*$^+$) astrocytes located in the SDH, an astrocyte subset that has an ability to induce pain sensitization. Furthermore, activation of *Hes5*$^+$ astrocytes reduced activity of SDH-inhibitory neurons (SDH-INs) that have an inhibitory role in pain processing. This astrocytic reduction of IN activity was canceled by an A$_1$-adenosine receptor (A$_1$R)-knockdown in SDH-INs, and the A$_1$R-knockdown suppressed pain hypersensitivity caused by acute restraint stress. Therefore, our findings suggest that LC$^{\rightarrow SDH}$-NA neuronal signaling to *Hes5*$^+$ SDH astrocytes and subsequent astrocytic reduction of SDH-IN activity are essential for mechanical pain facilitation caused by stress.

## Introduction

Somatosensory information generated at the periphery, such as skin, is conveyed to the dorsal horn of the spinal cord (SDH) via primary afferent sensory neurons and further to the brain (*Todd, 2010*; *Peirs and Seal, 2016*). The SDH is the first region for processing the peripherally derived sensory information and thus is critical for its proper transmission to and perception in the brain (*Todd, 2010*; *Peirs and Seal, 2016*; *Koch et al., 2018*; *Moehring et al., 2018*; *Peirs et al., 2020*; *Wang et al., 2022b*). The processing of somatosensory information is not only tightly regulated by complex neuronal circuits within the SDH but also powerfully modulated by descending neurons from the brain (*Mercer Lindsay et al., 2021*; *Nguyen et al., 2023*). The locus coeruleus (LC), a small nucleus located in the brainstem, modulates pain information processing in the SDH via descending noradrenaline (NA) neurons (*Nguyen et al., 2023*; *Yoshimura and Furue, 2006*; *Llorca-Torralba et al., 2016*; *Suárez-Pereira et al., 2022*). Activation of SDH-projecting LC-NA (LC$^{\rightarrow SDH}$-NA) neurons leads to a release of NA in the SDH (*Li et al., 2022*). Spinal NA acts on $\alpha_{1A}$-adrenaline receptors ($\alpha_{1A}$Rs) expressed on inhibitory neurons (INs) (*Häring et al., 2018*; *Shiraishi et al., 2021*) and facilitates $\gamma$-aminobutyric acid (GABA)-mediated inhibitory synaptic transmission (*Baba et al., 2000*; *Uchiyama et al., 2022*). This NA signal has been implicated in the inhibitory control of nociceptive information processing (*Yoshimura and Furue, 2006*; *Llorca-Torralba et al., 2016*; *Suárez-Pereira et al., 2022*). Besides neurons, glial cells in the SDH, such as microglia and astrocytes, also have a remarkable ability to modulate the processing and transmission of somatosensory information (*Inoue and Tsuda, 2018*; *Ji et al., 2019*; *Tsuda et al., 2023*). Glial modulation has been extensively studied in pathological settings (*Inoue and Tsuda, 2018*; *Ji et al., 2019*; *Tsuda et al., 2023*), but studies using in vivo imaging have shown that SDH astrocytes respond to noxious stimuli applied to the periphery under normal conditions (*Sekiguchi et al., 2016*; *Yoshihara et al., 2018*; *Kohro et al., 2020*; *Kawanabe et al., 2021*). Furthermore, we have recently identified a subset of spinal astrocytes defined by expression of hairy and enhancer of split 5 (*Hes5*) as a new target of NA (*Kohro et al., 2020*). Astrocytes express various types of adrenaline receptors (*Hertz et al., 2010*); activation of LC$^{\rightarrow SDH}$-NA neurons induces an increase in intracellular $Ca^{2+}$ ($[Ca^{2+}]_i$), an indicator of astrocyte activity (*Bazargani and Attwell, 2016*), via $\alpha_{1A}$Rs (*Kohro et al., 2020*), which are the most abundant adrenaline receptors in astrocytes (*Batiuk et al., 2020*; *Wahis et al., 2024*). *Hes5$^+$* astrocytes are selectively distributed in the superficial laminae of SDH where it receives primary afferent sensory fibers. Activation of $\alpha_{1A}$Rs of *Hes5$^+$* SDH astrocytes by intrathecal administration of NA induces pain sensitization to light mechanical stimuli applied to the skin (*Kohro et al., 2020*). Given that most of the varicosities of NAergic axons/terminals are located closer to astrocytic processes than to neuronal synapses (*Cohen et al., 1997*), *Hes5$^+$* astrocytes could be a critical NA target for modulating pain in the SDH.

LC-NA neurons are known to respond to several external and internal stressors (*Valentino and Van Bockstaele, 2008*). In rodent models of stress, nociceptive stimuli (e.g., electric shock, animal bite, etc.) are frequently employed as stressors. LC-NA neurons respond not only to stimuli that directly excite nociceptors (*Imbe et al., 2009*; *Chen and Sara, 2007*), but also to stressors that do not involve their direct stimulation (*Imbe et al., 2009*; *McCall et al., 2015*). Several studies using rodent stress models have shown that exposure to stressors with or without nociceptor activation induces a remarkable change in pain-associated behavior (*Butler and Finn, 2009*; *Jennings et al., 2014*). Despite the impact of stress on the pain system, the mechanism underlying a link between stress and pain, especially pain exacerbation, at the neural circuit and cellular levels remains poorly understood, although it has been proposed that activation of enkephalinergic and GABAergic INs in the SDH is involved in stress-induced pain inhibition (*François et al., 2017*). Since stressful events are known to have a significant impact on chronic pain (*Abdallah and Geha, 2017*; *Fitzcharles et al., 2021*), providing new insights into the mechanisms by which stress modulates pain is also of clinical importance.

In this study, we investigated how stress modulates pain with a focus on LC$^{\rightarrow SDH}$-NA neurons and their target *Hes5$^+$* astrocytes in the SDH, using multiple approaches. Our experiments using in vivo $Ca^{2+}$ imaging, chemogenetics and optogenetics, cell ablation, electrophysiology, biochemical imaging for NA release and astrocytic $Ca^{2+}$ responses, conditional gene knockout, and behavioral analyses demonstrate for the first time that LC$^{\rightarrow SDH}$-NA neuronal signaling to *Hes5$^+$* SDH astrocytes and subsequent astrocytic reduction of SDH-IN activity are essential for stress-induced mechanical pain hypersensitivity.

## Results

### LC$^{\rightarrow SDH}$-NA neurons mediate mechanical hypersensitivity after acute restraint stress

To investigate the mechanism underlying stress-induced pain modulation, we subjected wild-type (WT) mice to acute restraint stress exposure (*Figure 1A*), a well-known and frequently used stress model (*Valentino and Van Bockstaele, 2008*). Consistent with a previous study (*Sousa et al., 2018*), restraint stress for 30 min and 1 hr induced hypersensitivity to light mechanical stimuli (by applying von Frey filaments) to the plantar skin of the hindpaw (*Figure 1B*). However, with longer exposure (2 hr) to restraint stress, mechanical hypersensitivity was attenuated (*Figure 1—figure supplement 1*). In addition, mice subjected to 2 hr restraint stress displayed antinociceptive responses to noxious heat stimuli (the hot-plate test) (*Figure 1—figure supplement 2*), which is consistent with our previous finding (*Uchiyama et al., 2022*) as well as numerous other reports (*Supplementary file 1*). In contrast to restraint stress, forced swim stress, which has been reported to produce antinociceptive effects on noxious heat (*Contet et al., 2006*), did not change mechanical pain sensitivity (*Figure 1—figure supplement 2*). Thus, our data suggest that under our experimental conditions, single exposure of 1-hr restraint stress causes behavioral pain hypersensitivity to mechanical stimuli, although stress-induced modulation of pain is a complex phenomenon influenced by multiple factors, including the stress model, intensity, and duration, as well as the sensory modality used for behavioral testing.

Since 1 hr exposure to restraint stress has also been reported to increase plasma corticosterone levels (*Kim et al., 2018*), in subsequent experiments, we chose 1 hr as the exposure time to restraint stress. To analyze LC-NA neuronal activity during restraint stress, we performed fiber photometry using *Slc6a2-Cre* mice (Cre is expressed in NAergic neurons under the control of the promoter of NA transporter [NET]) (*Wagatsuma et al., 2018*) with microinjection into the LC of an adeno-associated viral (AAV) vector designed to express GCaMP6s (genetically encoded fluorescent $Ca^{2+}$ indicator) in a Cre-dependent manner (AAV-FLEx[GCaMP6s]). Three weeks after the injection, GCaMP6s was expressed selectively in tyrosine hydroxylase$^+$ (TH$^+$) LC-NA neurons (*Figure 1C*; *Mulvey et al., 2018*). We found that restraint stress exposure increased the number of $Ca^{2+}$ events in these neurons (*Figure 1D, E*; *Figure 1—figure supplement 3*; *Figure 1—video 1*). The number of $Ca^{2+}$ events was highest during the first 20 min of the restraint stress exposure and gradually decreased later.

To investigate the role of LC-NA neurons in mechanical hypersensitivity, we utilized a cell ablation approach (*Saito et al., 2001*); diphtheria toxin (DTX) receptors (DTR) were specifically expressed in LC-NA neurons by injection of AAV-FLEx[DTR-EGFP] into the LC of *Slc6a2-Cre* mice (*Figure 1F*). DTR$^+$ neurons (detected by GFP) mainly co-expressed TH, and these neurons were ablated by DTX administration (10 μg/kg, two injections 24 hr apart), which resulted in a loss of TH signal in the LC and reduction of NET signal in the SDH (*Figure 1G*). We performed behavioral tests in these mice and observed that ablation of LC-NA neurons abolished mechanical hypersensitivity after restraint stress exposure (*Figure 1H*). To specifically ablate LC$^{\rightarrow SDH}$-NA neurons, a retrograde AAV vector expressing Cre (AAVretro-Cre) was injected into the L4 SDH, followed by AAV-FLEx[DTR-EGFP] injection into the LC (*Figure 1I*). In these mice, TH$^+$ LC$^{\rightarrow SDH}$-NA neurons expressing DTR (detected by EGFP) were ablated after DTX administration (*Figure 1J*; *Figure 1—figure supplement 4A, B*) (4.4% of total TH$^+$ neurons in the LC (189 DTR$^+$ cells per 4398 TH$^+$ cells, *n* = 3 mice)). DTR$^+$ cells were not observed in A5 or A7 (*Figure 1—figure supplement 4C*), confirming the specificity of LC$^{\rightarrow SDH}$-NAergic pathway targeting in our study. These mice also failed to induce mechanical hypersensitivity after the restraint stress (*Figure 1K*). These results suggest that descending LC$^{\rightarrow SDH}$-NA neurons are necessary for mechanical pain hypersensitivity after acute restraint stress exposure. In contrast, motor behavior in the rotarod test and behavioral sensitivity to heat stimuli in the paw-flick test were not affected by ablation of LC$^{\rightarrow SDH}$-NA neurons (*Figure 1—figure supplement 5*), indicating that ablation of these neurons does not produce non-specific behavioral deficits in motor function or other sensory modalities. Furthermore, we tested whether optogenetic activation of LC$^{\rightarrow SDH}$-NA neurons could induce mechanical hypersensitivity, using ChrimsonR, a well-established red-shifted channelrhodopsin (*Klapoetke et al., 2014*). *Slc6a2-Cre* mice were microinjected with AAV-FLEx[ChrimsonR-tdTomato] into the LC (*Figure 1L*). Three weeks later, we confirmed expression of ChrimsonR (detected by tdTomato) in TH$^+$ neurons in the LC and NET$^+$ axons/terminals in the SDH (*Figure 1M*; *Figure 1—figure supplement 6*). Optogenetic stimulation of LC$^{\rightarrow SDH}$-NA neuronal axons/terminals in the SDH evoked mechanical hypersensitivity in these mice (*Figure 1N*). These results together demonstrate that LC$^{\rightarrow SDH}$-NA neurons are

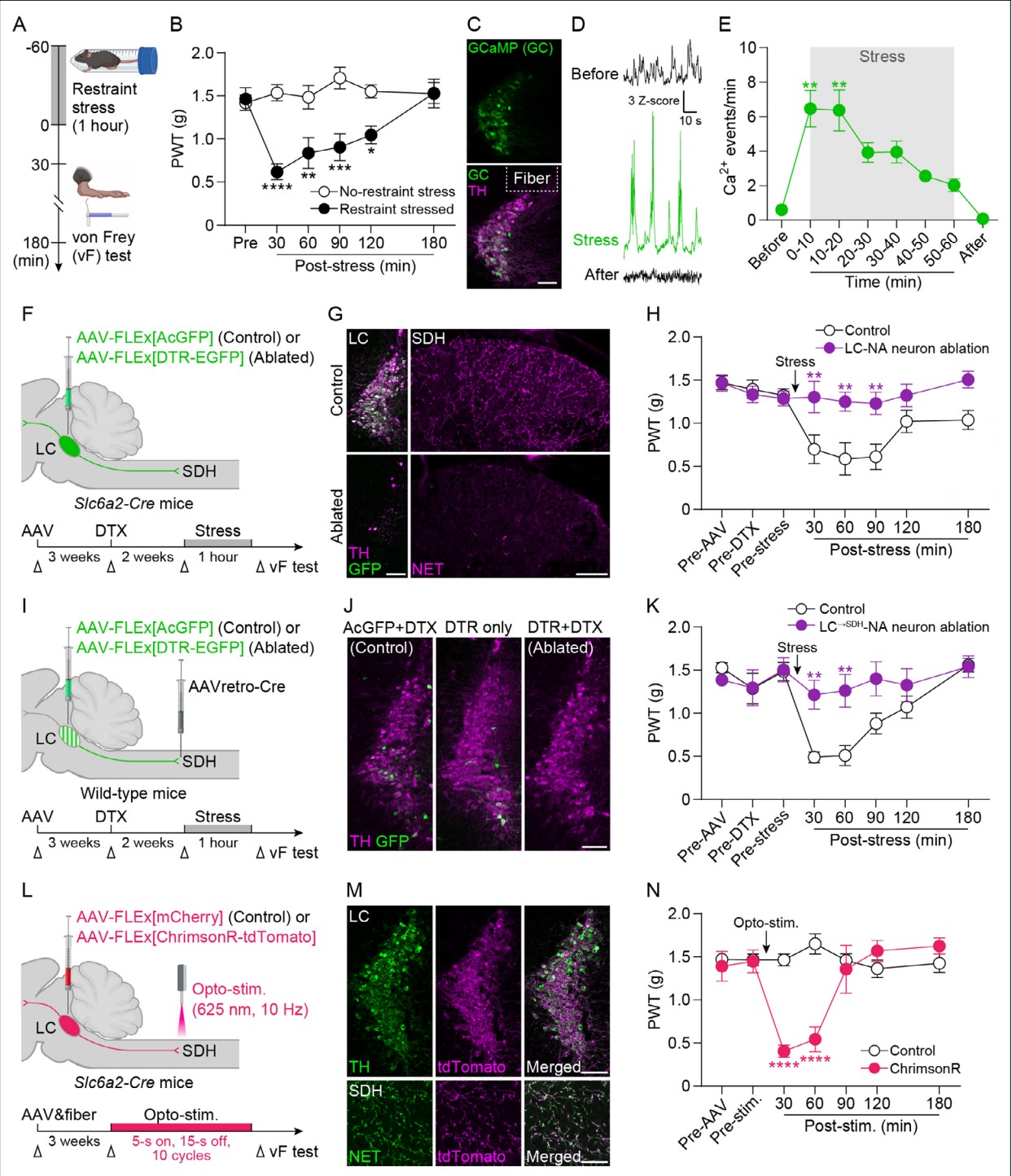

**Figure 1.** LC→SDH-NA neurons mediate stress-induced mechanical hypersensitivity. (**A**) Schematic illustration of an experiment to investigate the effects of acute exposure to restraint stress (1 hr) on mechanosensory behavior in mice, using von Frey (vF) filaments. (**B**) Change in paw withdrawal threshold (PWT) measured by vF filaments in wild-type mice after restraint stress ($n$ = 6 mice per group; two-way ANOVA with post hoc Bonferroni's multiple comparisons test; *$p < 0.05$, **$p < 0.01$, ***$p < 0.001$, ****$p < 0.0001$ vs. no-restraint stress group). (**C**) Expression of GCaMP6s (GC; green) in the LC at 3 weeks after intra-LC injection of AAV-FLEx[GCaMP6s] in *Slc6a2-Cre* mice. TH immunofluorescence is shown in magenta. Dashed line indicates the location of the implanted optic fiber. Scale bar, 100 μm. (**D, E**) Representative traces and change in the frequency of GCaMP6s signals in LC-NA neurons ($n$ = 6 mice; Friedman test with post hoc Dunn's multiple comparisons test; **$p < 0.01$ vs. the data of 'Before'). Traces shown at the top, middle, and bottom (**D**) indicate $Ca^{2+}$ signals before, during, and after restraint stress, respectively. (**F**) Schematic illustration of the strategy of ablating LC-NA neurons using AAV vectors incorporating DTR (fused with EGFP) injected into the LC in *Slc6a2-Cre* mice. (**G**) TH immunofluorescence

*Figure 1 continued on next page*

*Figure 1 continued*

(magenta) and GFP (green) in the LC (left) and NET immunofluorescence (magenta) in the SDH (right) after administration of DTX (10 µg/kg, i.p., two injections 24 hr apart) in control mice (top) and DTR-expressing mice (bottom). Scale bars, 100 µm. (**H**) Effect of ablation of LC-NA neurons on PWT changes after acute restraint stress ($n$ = 7 mice per group; two-way ANOVA with post hoc Bonferroni's multiple comparisons test; **p < 0.01 vs. control group). (**I**) Schematic illustration of the strategy of ablating $LC^{\to SDH}$-NA neurons using a retrograde AAV vector incorporating Cre injected into the SDH and an AAV vector incorporating DTR (fused with EGFP) injected into the LC in wild-type mice. (**J**) Representative images of $LC^{\to SDH}$-NA neurons in control or DTR-expressing mice treated with vehicle or DTX administration, respectively. GFP (green) and TH (magenta). Scale bar, 100 µm. (**K**) Effect of ablation of $LC^{\to SDH}$-NA neurons on PWT changes after restraint stress ($n$ = 11 mice per group; two-way ANOVA with post hoc Bonferroni's multiple comparisons test; **$P$<0.01 vs. control group). (**L**) Schematic illustration of the strategy for activating $LC^{\to SDH}$-NA neuronal axons/terminals using an AAV vector incorporating ChrimsonR (fused with tdTomato) injected into the LC in *Slc6a2-Cre* mice and of an optic cannula implanted in the SDH. (**M**) Representative images of TH (green) and tdTomato (magenta) expression in the LC (top) and NET (green) and tdTomato (magenta) expression in the SDH (bottom) at 3 weeks after intra-LC injection of AAV-FLEx[ChrimsonR-tdTomato] in *Slc6a2-Cre* mice. Scale bars, 100 µm (top) and 50 µm (bottom). (**N**) PWT before and after optogenetic stimulation (opto-stim.) in $LC^{\to SDH}$-NA axons/terminals (625 nm, 2 mW, 10 Hz, 5 ms pulse duration, 5 s light on, 15 s light off, 10 cycles) (Control, $n$ = 4 mice; ChrimsonR, $n$ = 5 mice; two-way ANOVA with post hoc Bonferroni's multiple comparisons test; ****p < 0.0001 vs. control group). Data represent mean ± SEM. See also *Figure 1—figure supplements 1–6*. Some figure elements were created with BioRender.com.

The online version of this article includes the following video, source data, and figure supplement(s) for figure 1:

**Source data 1.** Raw numerical values for *Figure 1* plots.

**Figure supplement 1.** Mechanical hypersensitivity is weak in mice with longer exposure (2 hr) to restraint stress.

**Figure supplement 1—source data 1.** Raw numerical values for *Figure 1—figure supplement 1* plots.

**Figure supplement 2.** Behavioral responses to thermal and mechanical stimuli in stress models.

**Figure supplement 2—source data 1.** Raw numerical values for *Figure 1—figure supplement 2* plots.

**Figure supplement 3.** Raw fluorescent signals in LC-NA neurons during restraint stress (in vivo fiber photometry).

**Figure supplement 4.** Specific expression and ablation of $LC^{\to SDH}$-NA neurons.

**Figure supplement 4—source data 1.** Raw numerical values for *Figure 1—figure supplement 4* plots.

**Figure supplement 5.** Ablation of $LC^{\to SDH}$-NA neurons does not affect motor function or thermal sensation.

**Figure supplement 5—source data 1.** Raw numerical values for *Figure 1—figure supplement 5* plots.

**Figure supplement 6.** Expression of ChrimsonR in the superficial and deeper SDH.

**Figure 1—video 1.** In vivo Ca²⁺ imaging of LC neurons in mice using fiber photometry during acute restraint stress.
https://elifesciences.org/articles/104453/figures#fig1video1

activated by acute stress exposure and are not only necessary for stress-induced mechanical pain hypersensitivity but also sufficient to phenocopy the behavioral response.

## $LC^{\to SDH}$-NAergic signaling to *Hes5*⁺ SDH astrocytes is required for stress-induced mechanical hypersensitivity

To investigate the mechanism by which activation of $LC^{\to SDH}$-NA neurons induces mechanical hypersensitivity, we first examined if NA is released in the SDH when these neurons are stimulated, using the genetically encoded fluorescent NA sensor $GRAB_{NE1m}$ (*Feng et al., 2019*). In spinal cord slices from *Slc6a2-Cre* mice that had been injected AAV-FLEx[ChrimsonR-tdTomato] into the LC and AAV-gfaABC₁D-$GRAB_{NE1m}$ (expressing $GRAB_{NE1m}$ under the control of the astrocyte-selective promoter gfaABC₁D (*Lee et al., 2008*) into the SDH) (*Figure 2A*), optogenetic stimulation of $LC^{\to SDH}$-NA axons/terminals increased $GRAB_{NE1m}$ fluorescence intensity (*Figure 2B*), confirming NA release from $LC^{\to SDH}$-NA neuron terminals in our slice conditions. The fluorescence intensity increased in a stimulation period-dependent manner (*Figure 2C*). Next, to determine the target cells and receptors of $LC^{\to SDH}$-NA neurons, we focused on $\alpha_{1A}$Rs in astrocytes because our previous study showed that NA increases $[Ca^{2+}]_i$ in astrocytes via $\alpha_{1A}$Rs (*Kohro et al., 2020*). Using spinal cord slices from *Slc6a2-Cre* mice that had been injected with AAV-FLEx[ChrimsonR-tdTomato] into the LC and AAV-gfaABC₁D-GCaMP6m into the SDH (*Figure 2A*), we found that optogenetic stimulation of $LC^{\to SDH}$-NA axons/terminals evoked a rise in $[Ca^{2+}]_i$ in astrocytes (*Figure 2D, E*). Pretreatment with the $\alpha_{1A}$R-specific antagonist silodosin (40 nM) suppressed the Ca²⁺ responses (*Figure 2F*), suggesting that NA released from $LC^{\to SDH}$-NA neurons induces Ca²⁺ responses in SDH astrocytes via $\alpha_{1A}$Rs. Behaviorally, intrathecal injection of silodosin in *Slc6a2-Cre*;AAV-ChrimsonR mice attenuated mechanical hypersensitivity by optogenetic stimulation of the $LC^{\to SDH}$-NA neurons (*Figure 2G*). Consistent with our previous findings (*Kohro*

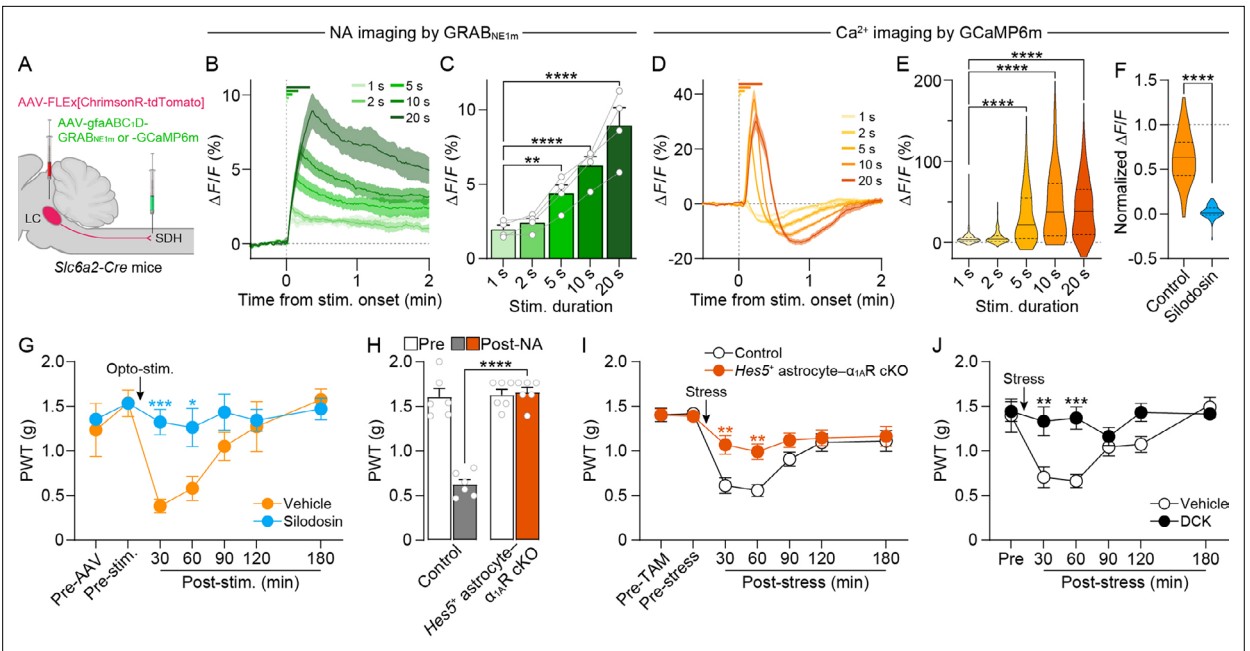

**Figure 2.** $\alpha_{1A}$Rs in $Hes5^+$ SDH astrocytes are required for stress-induced mechanical hypersensitivity. (**A**) Schematic illustration of intra-SDH microinjection of AAV-gfaABC₁D-GRAB$_{NE1m}$ or -GCaMP6m and intra-LC microinjection of AAV-FLEx[ChrimsonR-tdTomato] in *Slc6a2-Cre* mice. (**B**) Representative traces of GRAB$_{NE1m}$ signals by fluorescence imaging using spinal cord slices. Each trace represents the GRAB$_{NE1m}$ signal before and after optogenetic stimulation (625 nm, 1 mW, 10 Hz, 5ms pulse duration, 1–20 s). (**C**) Quantitative analysis of the peak amplitude of GRAB$_{NE1m}$ $\Delta F/F$ after optogenetic stimulation in LC$^{\to SDH}$-NA axons/terminals ($n$ = 4 slices; one-way ANOVA with post hoc Dunnett's multiple comparisons test; **$p < 0.01$, ****$p < 0.0001$). (**D**) Representative traces of astrocytic GCaMP6m signals by fluorescence imaging using spinal cord slices. Each trace represents the GCaMP6m signal before and after optogenetic stimulation (as described in B). (**E**) Quantitative analysis of the peak amplitude of GCaMP6m $\Delta F/F$ after optogenetic stimulation in LC$^{\to SDH}$-NA axons/terminals ($n$ = 133 cells, 4 slices, 4 mice; Friedman test with post hoc Dunn's multiple comparisons test; ****$p < 0.0001$). (**F**) Effect of silodosin (40 nM) on astrocytic Ca$^{2+}$ responses in the SDH after optogenetic stimulation (10 s) in LC$^{\to SDH}$-NA axons/terminals (Control, $n$ = 83 cells, 4 slices, 4 mice; Silodosin, $n$ = 53 cells, 4 slices, 4 mice; Mann–Whitney test; ****$p < 0.0001$). (**G**) Effect of intrathecal silodosin (3 nmol) on mechanical hypersensitivity induced by optogenetic stimulation in LC$^{\to SDH}$-NA axons/terminals (Vehicle, $n$ = 5 mice; Silodosin, $n$ = 6 mice; two-way ANOVA with post hoc Bonferroni's multiple comparisons test; *$p < 0.05$, ***$p < 0.001$ vs. vehicle group). (**H**) Change in PWT at 30 min after intrathecal injection of NA (0.1 nmol) in control (*Adra1a*$^{flox/flox}$) and $Hes5^+$ astrocyte-selective $\alpha_{1A}$R conditional knockout mice *Hes5-CreERT2;Adra1a*$^{flox/flox}$ mice treated with tamoxifen (TAM) (*Hes5$^+$ astrocyte–$\alpha_{1A}$R cKO* mice) ($n$ = 6 mice per group; two-way ANOVA with post hoc Bonferroni's multiple comparisons test; ****$p < 0.0001$). (**I, J**) Stress-induced mechanical hypersensitivity in $Hes5^+$ astrocyte–$\alpha_{1A}$R cKO mice [I: Control (*Adra1a*$^{flox/flox}$), $n$ = 7 mice; $Hes5^+$ astrocyte–$\alpha_{1A}$R cKO, $n$ = 8 mice] or wild-type mice with intrathecal DCK (10 nmol) (**J**: $n$ = 5 mice per group) (two-way ANOVA with post hoc Bonferroni's multiple comparisons test; **$p < 0.01$, ***$p < 0.001$ vs. control or vehicle group). Data represent mean ± SEM. Some figure elements were created with BioRender.com.

The online version of this article includes the following source data for figure 2:

**Source data 1.** Raw numerical values for *Figure 2* plots.

et al., 2020), intrathecal administration of NA (0.1 nmol) evoked transient mechanical hypersensitivity, which was completely abolished in *Hes5-CreERT2;Adra1a*$^{flox/flox}$ mice treated with tamoxifen (*Hes5$^+$ astrocyte-selective conditional knockout of $\alpha_{1A}$Rs (Kohro et al., 2020): Hes5$^+$ astrocyte–$\alpha_{1A}$R cKO*) (***Figure 2H***). Moreover, mechanical hypersensitivity after acute restraint stress was abolished in $Hes5^+$ astrocyte–$\alpha_{1A}$R cKO mice (***Figure 2I***). In addition, intrathecal administration of 5,7-dichlorokynurenic acid (DCK; 10 nmol), an antagonist of D-serine signaling on *N*-methyl-D-aspartate (NMDA) receptors, also suppressed mechanical hypersensitivity (***Figure 2J***). Supporting this finding, our previous report identified D-serine as a hypersensitivity-inducing factor released from $Hes5^+$ astrocytes (***Kohro et al., 2020***). These results indicate that spinally released NA from LC$^{\to SDH}$-NA neurons after acute restraint stress activates $\alpha_{1A}$Rs on $Hes5^+$ astrocytes, which causes mechanical pain hypersensitivity.

## $Hes5^+$ astrocytes mask IN function via adenosine A₁ receptors

Besides astrocytes, *Adra1a* mRNA is also expressed in INs in the SDH (***Häring et al., 2018; Shiraishi et al., 2021***). However, *Slc32a1-Cre;Adra1a*$^{flox/flox}$ mice (*Slc32a1$^+$* (the gene that encodes the vesicular

GABA transporter) IN-selective conditional knockout of $\alpha_{1A}$Rs: $Slc32a1^+$ IN–$\alpha_{1A}$R cKO mice) (*Shiraishi et al., 2021*; *Uchiyama et al., 2022*) had no effect on the NA (0.1 nmol)-induced mechanical hypersensitivity (*Figure 3A*). A similar behavioral phenotype was also observed after intrathecal administration of phenylephrine (0.05 nmol), a selective agonist for $\alpha_1$Rs (*Figure 3B*). Further-more, stress-induced mechanical hypersensitivity was not affected in $Slc32a1^+$ IN–$\alpha_{1A}$R cKO mice (*Figure 3C*). Considering previous data that SDH-INs (including $Slc32a1^+$) respond to NA via $\alpha_{1A}$Rs at NA concentrations similar to those that elicit $Hes5^+$ astrocyte response (*Uchiyama et al., 2022*; *Kohro et al., 2020*), it remains unclear why intrathecally administered NA and spinally released NA from $LC^{\rightarrow SDH}$-NA neuron terminals (after acute restraint stress) have no suppressive effect via $\alpha_{1A}$R-mediated $Slc32a1^+$ IN activation on mechanical hypersensitivity. We then hypothesized an interaction between $Hes5^+$ astrocytes and $Slc32a1^+$ INs as per a previous report that optogenetic activation of rat SDH astrocytes leads to ATP release; ATP is converted to adenosine extracellularly, which subse-quently suppresses the activity of SDH-INs via adenosine $A_1$ receptors ($A_1$Rs) (*Nam et al., 2016*). By performing whole-cell current-clamp recordings of $Slc32a1^+$ INs in spinal cord slices from *Slc32a1-Cre;Rosa26-LSL-tdTomato* mice, we found that bath application of NA (20 µM) evoked depolarization in 22.2% (10/45 cells) of $tdTomato^+$ cells ($Slc32a1^+$ INs), with 30% of these showing action potentials (3/10 cells) (*Figure 3D*). In our experiments, four antagonists for AMPA and NMDA receptors and $GABA_A$ and glycine receptors were added to the aCSF to block synaptic inputs from other neurons to the recorded $Slc32a1^+$ neurons (*Wu et al., 2004*; *Liu et al., 2010*), suggesting that NA directly acts on the recorded SDH $Slc32a1^+$ neurons to produce excitation. The proportion of $Slc32a1^+$ INs depolarized by NA is similar to that of *Adra1a* mRNA-expressing INs (*Shiraishi et al., 2021*). In 50% (5/10 cells) of depolarized $Slc32a1^+$ INs, additional application of the $A_1$R-selective agonist N6-cyclo-pentyladenosine (CPA; 1 µM) suppressed NA-induced depolarization/action potentials (*Figure 3D, E*). Supporting this finding, RNAscope in situ hybridization detected *Adora1* mRNA fluorescence in 63.5 ± 6.9% of $Slc32a1^+$ INs ($n$ = 497, from three mice) (*Figure 3F*). Because $A_1$Rs are also expressed in the dorsal root ganglion neurons (*Metzner et al., 2021*), excitatory neurons and microglia (*Cui et al., 2024*) in the SDH, we employed the CRISPR–Cas9 system using AAV vectors to specifically knockdown $A_1$Rs in SDH-INs. AAV vectors designed to express the *Staphylococcus aureus* Cas9 (SaCas9) (*Ran et al., 2015*) in a Cre-dependent manner (AAV-FLEx[SaCas9]) and single guide RNA-expression vectors (AAV-FLEx[mCherry]-U6-sgAdora1 or -FLEx[mCherry]-U6-sgYFP [as a control]) were co-microinjected into the SDH of *Slc32a1-Cre* mice (*Figure 3G*). SaCas9 (detected by HA-tag) and mCherry-labeled cells in the SDH were immunolabeled with paired box 2 (PAX2) (*Figure 3G*), a marker of INs (*Cheng et al., 2004*). In these SDH-$Slc32a1^+$ IN-selective $A_1$R conditional knockdown mice (SDH-$Slc32a1^+$ IN–$A_1$R cKD mice), the inhibitory effect of CPA on NA-induced depolarization/action potentials was completely abolished in $mCherry^+$ neurons (*Figure 3H, I*). Using a chemo-genetic approach with modified human muscarinic $G_q$-protein-coupled receptors (hM3Dq) (*Roth, 2016*), we further examined the effect of $G_q$-stimulated $Hes5^+$ astrocytes on inhibitory synaptic trans-mission. *Hes5-CreERT2* mice were microinjected with AAV-FLEx[hM3Dq-HA] into the SDH (*Hes5-CreERT2;AAV-hM3Dq* mice). hM3Dq (detected by HA-tag) was expressed in the SDH, and these cells were double-labeled with glial fibrillary acidic protein (GFAP) and SRY-related high-mobility group box 9 (SOX9), markers of astrocytes (*Figure 3J*). Our previous studies using *Hes5-CreERT2* mice have confirmed that hM3Dq is not expressed in other cell types (neurons, oligodendrocytes, or microglia) (*Kohro et al., 2020*; *Kagiyama et al., 2025*). Using spinal cord slices from these mice, we performed whole-cell voltage-clamp recordings of substantia gelatinosa (SG) neurons. Bath appli-cation of hM3Dq agonist clozapine N-oxide (CNO; 100 µM) reduced the frequency of spontaneous inhibitory postsynaptic currents (sIPSCs) in SG neurons (*Figure 3K, L*). The concentration of CNO has been used in our previous study (*Kohro et al., 2020*) but is relatively high; however, 100 µM CNO most effectively replicated the NA-induced $Ca^{2+}$ signals in astrocytes (*Figure 3—figure supplement 1A and B*). In addition, spinal cord slices from mice that did not express hM3Dq, 100 µM CNO had no effect on $Ca^{2+}$ responses in SDH astrocytes (*Figure 3—figure supplement 1A*) and sIPSCs in SG neurons (*Figure 3—figure supplement 1C*), confirming that the observed effect of CNO is not non-specific. The reduction of sIPSCs by astrocytic hM3Dq activation was prevented by pretreatment with the $A_1$R-specific antagonist, 8-cyclopentyl-1,3-dimethylxanthine (CPT; 1 µM) (*Figure 3M, N*). These results suggest that a signaling pathway from $Hes5^+$ astrocytes suppresses the activity of $Slc32a1^+$ INs via $A_1$Rs in the SDH.

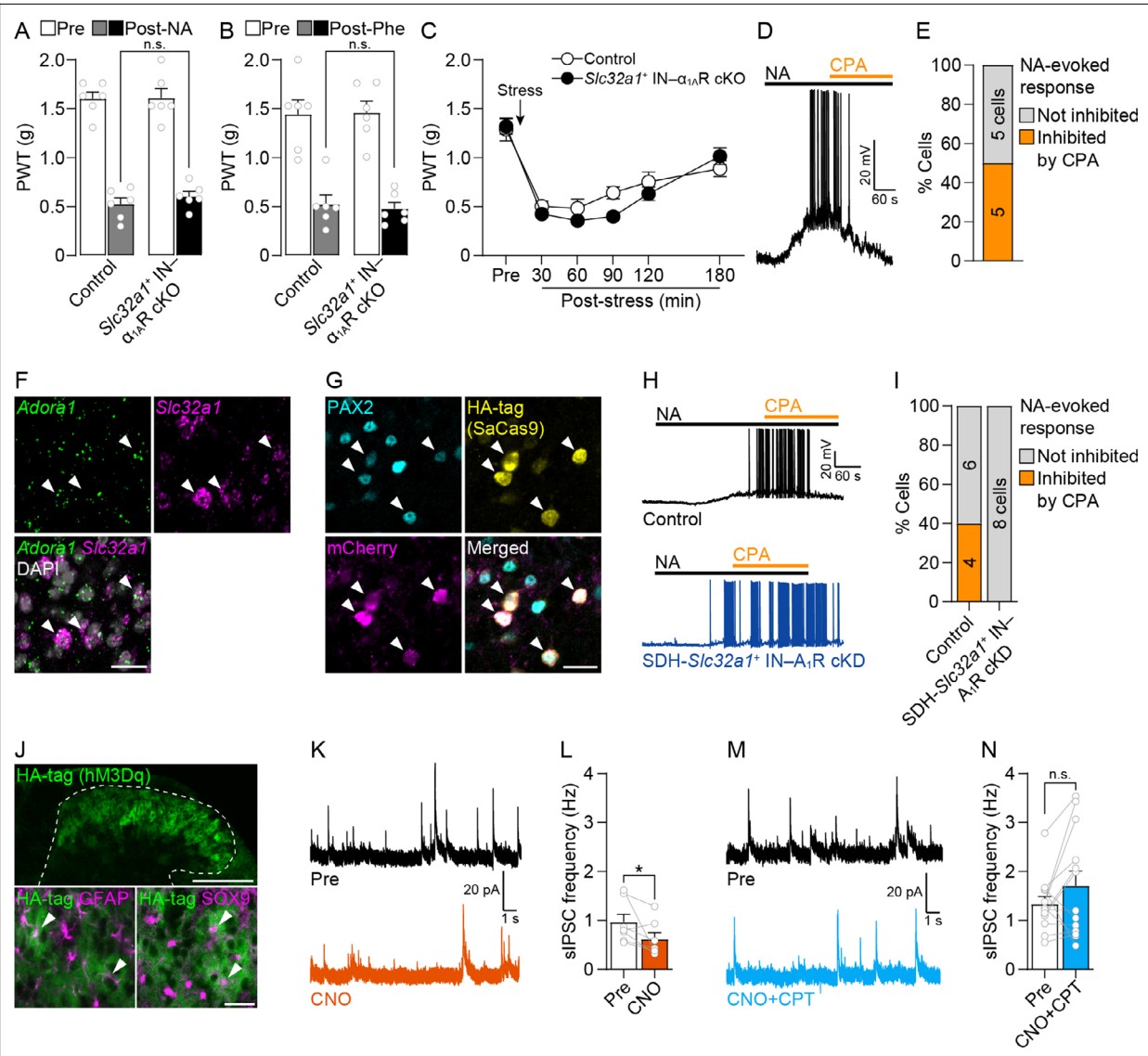

**Figure 3.** Activation of *Hes5*[+] astrocytes reduces activity in *Slc32a1*[+] INs in the SDH. Intrathecal NA (**A**, 0.1 nmol) or Phe (**B**, 0.05 nmol)-induced mechanical hypersensitivity in control (*Adra1a*[flox/flox]) and *Slc32a1*[+] IN-selective $\alpha_{1A}$R conditional knockout mice [*Slc32a1-Cre;Adra1a*[flox/flox] mice (*Slc32a1*[+] INs–$\alpha_{1A}$R cKO mice)] (*n* = 6 mice per group; two-way ANOVA with post hoc Bonferroni's multiple comparisons test; n.s., not significant). (**C**) Changes in PWT after acute exposure to restraint stress in control (*Adra1a*[flox/flox]; *n* = 6 mice) and *Slc32a1*[+] INs–$\alpha_{1A}$R cKO mice (*n* = 9 mice) (two-way ANOVA with post hoc Bonferroni's multiple comparisons test). (**D**) Representative trace of membrane potentials in tdTomato[+] (*Slc32a1*[+]) SDH neurons after application of NA (20 µM) to spinal cord slices from *Slc32a1-Cre;Rosa26-LSL-tdTomato* mice. $A_1$R agonist CPA (1 µM) was co-applied with NA. (**E**) Percentage of *Slc32a1*[+] SDH neurons whose NA-evoked response was inhibited by CPA (*n* = 10 cells from 7 mice). (**F**) Representative images of *Adora1* ($A_1$R) mRNA expression (green) in *Slc32a1*[+] INs (magenta). DAPI staining is shown in gray. Arrowheads indicate $A_1$R-expressing *Slc32a1*[+] cells. Scale bar, 25 µm. (**G**) SaCas9 (yellow, detected by HA-tag) and mCherry (magenta) expression in the PAX2[+] INs (cyan) at 3 weeks after intra-SDH injection of AAV-FLEx[SaCas9] and AAV-FLEx[mCherry]-U6-sgAdora1 in *Slc32a1-Cre* mice. Arrowheads indicate genome-editing *Slc32a1*[+] cells. Scale bar, 25 µm. (**H**) Representative traces of membrane potentials in *Slc32a1*[+] INs after application of NA and CPA to spinal cord slices from *Slc32a1-Cre* mice with conditional knockdown of $A_1$Rs in *Slc32a1*[+] INs (SDH-*Slc32a1*[+] IN–$A_1$R cKD) and their controls (Control; *Slc32a1-Cre* mice with intra-SDH injection of AAV-FLEx[mCherry]). (**I**) Percentage of mCherry[+] (*Slc32a1*[+]) SDH neurons whose NA-evoked response was inhibited by CPA (Control, *n* = 10 cells from 8 mice; SDH-*Slc32a1*[+] IN–$A_1$R cKD, *n* = 8 cells from 5 mice). (**J**) Expression of hM3Dq (green, detected by HA-tag) in the SDH at 3 weeks after intra-SDH injection of AAV-FLEx[hM3Dq] in *Hes5-CreERT2* mice treated with TAM. Dashed line indicates an outline of the gray matter of SDH. GFAP (magenta, bottom left), SOX9 (magenta, bottom right). Arrowheads indicate HA-tag[+] astrocytes. Scale bars, 200 µm (top) and 25 µm (bottom). Representative traces of spontaneous inhibitory postsynaptic currents (sIPSCs) (**K**) and quantitative analysis of their frequency (**L**) in SG neurons in spinal cord slices from *Hes5-CreERT2;AAV-hM3Dq* mice treated with TAM [Pre and CNO: before and after bath application of CNO (100 µM), respectively] (*n* = 7 cells from 7 mice; Wilcoxon signed-rank test; *p < 0.05). Representative traces of sIPSCs (**M**) and quantitative analysis of their frequency (**N**) in SG neurons in spinal cord slices from *Hes5-CreERT2;AAV-FLEx[hM3Dq]* mice treated with TAM [Pre and CNO: before and after bath application of CNO with CPT

*Figure 3 continued on next page*

*Figure 3 continued*

(1 µM), respectively] (*n* = 13 cells from 13 mice; Wilcoxon signed-rank test; n.s., not significant). Data represent mean ± SEM. See also ***Figure 3—figure supplements 1 and 2***.

The online version of this article includes the following source data and figure supplement(s) for figure 3:

**Source data 1.** Raw numerical values for ***Figure 3*** plots.

**Figure supplement 1.** Effects of CNO on Ca$^{2+}$ response in astrocytes and IPSCs in SG neurons in spinal cord slices from mice with or without hM3Dq expression.

**Figure supplement 1—source data 1.** Raw numerical values for ***Figure 3—figure supplement 1*** plots.

**Figure supplement 2.** Knockdown of A$_1$Rs in *Slc32a1*$^+$ neurons tends to increase the proportion of *Slc32a1*$^+$ neurons with NA-induced depolarizing responses.

**Figure supplement 2—source data 1.** Raw numerical values for ***Figure 3—figure supplement 2*** plots.

## *Hes5*$^+$ astrocyte-mediated inhibitory signals to INs affect mechanical hypersensitivity

We investigated whether A$_1$Rs in SDH-*Slc32a1*$^+$ INs contributed to the mechanical hypersensitivity elicited by intrathecal NA (0.1 nmol). Acute inhibition of spinal A$_1$Rs by intrathecal administration of CPT (3 nmol) suppressed NA-induced mechanical hypersensitivity (***Figure 4A***). Similarly, SDH-*Slc32a1*$^+$ IN–A$_1$R cKD mice showed a significant reduction in the behavioral hypersensitivity by NA (***Figure 4B***), indicating that A$_1$Rs in SDH-*Slc32a1*$^+$ INs contribute to mechanical hypersensitivity by spinal NA. Furthermore, SDH-*Slc32a1*$^+$ IN–A$_1$R cKD mice and intrathecal-CPT-pretreated mice exhibited a significant attenuation of the stress-induced mechanical hypersensitivity (***Figure 4C, D***). These results suggest that *Hes5*$^+$ astrocyte-mediated negative control for SDH-*Slc32a1*$^+$ INs via A$_1$R-mediated signals is crucial for stress-induced mechanical hypersensitivity. Moreover, we investigated the role of this interaction in stress-induced alteration of somatosensory information processing in the SDH using c-FOS, a neuronal activity marker, and phosphorylated extracellular signal-regulated kinase (pERK), a neuronal nociceptive input marker in superficial SDH neurons (***Ji et al., 1999***). Consistent with the results of a previous report (***Matsumoto et al., 2008***), Aβ fiber stimulation of the hindpaw alone did not increase the number of c-FOS$^+$ (***Figure 4E, F***) and pERK$^+$ neurons (***Figure 4G, H***) in the superficial SDH (laminae I–IIi using isolectin B4 [IB4], a marker of lamina IIi). Similarly, acute restraint stress alone did not change the number of c-FOS$^+$ and pERK$^+$ neurons (***Figure 4E–H***). Interestingly, Aβ fiber stimulation following acute restraint stress exposure significantly increased the number of c-FOS$^+$ and pERK$^+$ neurons in the superficial SDH (***Figure 4E–H***). Intrathecal injection of CPT prevented the stress-induced increase in c-FOS$^+$ and pERK$^+$ neurons (***Figure 4E–H***). These results suggest that the abnormal responsiveness of superficial SDH neurons to signals from Aβ fibers in mice with acute restraint stress involves adenosine signals that are important for negatively controlling *Slc32a1*$^+$ INs by NA-stimulated *Hes5*$^+$ astrocytes and for mechanical hypersensitivity.

## Discussion

External and internal stressors elicit a variety of biological responses including pain modulation. Acute exposure to a stressor inhibits or facilitates behavioral responses to thermal and mechanical stimuli applied to the skin (***Butler and Finn, 2009***; ***Sousa et al., 2018***; ***Kawabata et al., 2023***). While the mechanism underlying pain inhibition has been studied (***Butler and Finn, 2009***), the neural basis of pain facilitation by stress remains poorly understood. In this study, we identified a circuit, the descending LC$^{\rightarrow SDH}$-NA neurons, that is activated by acute restraint stress exposure and demonstrated that the circuit is necessary for stress-induced mechanical pain hypersensitivity. A notable finding of this study was that *Hes5*$^+$ astrocytes, a subpopulation of spinal cord astrocytes (***Kohro et al., 2020***), are the primary target of NA released from descending LC$^{\rightarrow SDH}$-NA neurons for mechanical pain hypersensitivity caused by stress. Thus, this top–down signaling pathway from LC$^{\rightarrow SDH}$-NA neurons to *Hes5*$^+$ SDH astrocytes is an essential link between stress and mechanical pain sensitization.

Spinal NA derived from LC-NA neurons has been classically implicated in pain inhibitory control (***Yoshimura and Furue, 2006***; ***Llorca-Torralba et al., 2016***; ***Suárez-Pereira et al., 2022***). The suggested mechanism to this effect is its action on SDH-INs via α$_{1A}$Rs (***Yoshimura and Furue, 2006***; ***Uchiyama et al., 2022***). In contrast, our recent study has unveiled another facet of spinal NA

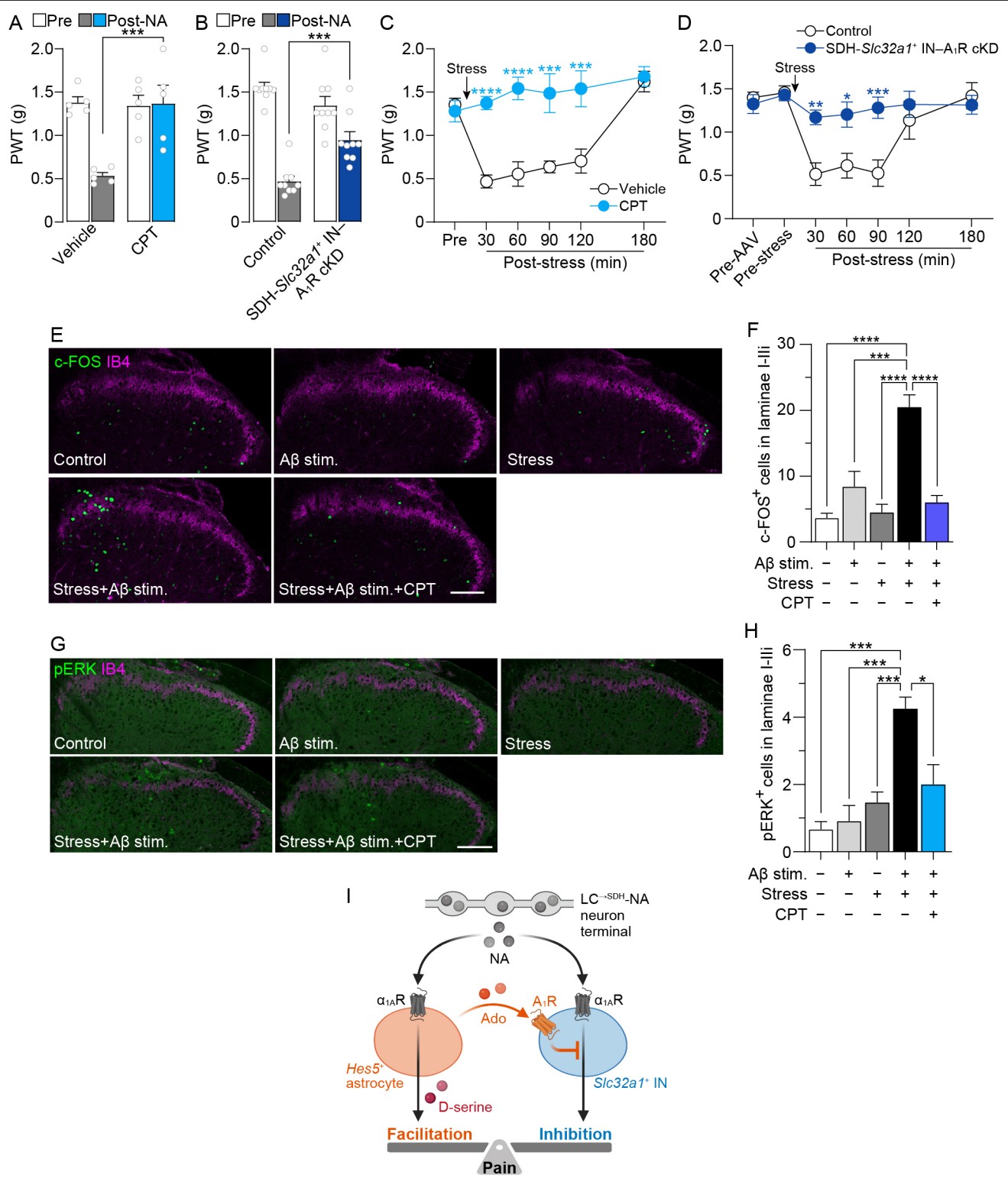

**Figure 4.** *Hes5+* astrocyte-mediated inhibitory signals to SDH-*Slc32a1+* INs contribute to stress-induced mechanical hypersensitivity. PWT before and 30 min after intrathecal administration of NA (0.1 nmol) in wild-type mice pretreated intrathecally with vehicle or CPT (3 nmol) (A: *n* = 5 mice per group) or in control (*Slc32a1-Cre* mice with intra-SDH of AAV-FLEx[mCherry]) and SDH-*Slc32a1+* IN–$A_1$R cKD mice (B: *n* = 9 mice per group) (two-way ANOVA with post hoc Bonferroni's multiple comparisons test; ***p < 0.001). PWT before and after acute restraint stress in wild-type mice pretreated intrathecally with vehicle or CPT (C: Vehicle, *n* = 6 mice; CPT, *n* = 5 mice) or in control (*Slc32a1-Cre* mice with intra-SDH of AAV-FLEx[mCherry]) and SDH-*Slc32a1+* IN–$A_1$R cKD mice (D: Control, *n* = 6 mice; SDH-*Slc32a1+* IN–$A_1$R cKD, *n* = 7 mice) (two-way ANOVA with post hoc Bonferroni's multiple comparisons test;

*Figure 4 continued on next page*

*Figure 4 continued*

*p < 0.05, **p < 0.01, ***p < 0.001, ****p < 0.0001 vs. vehicle or control group). (**E–H**) Representative images of c-FOS (green, **E**), pERK (green, **G**), and IB4 (magenta, **E**, **G**) immunofluorescence in the SDH with or without Aβ fiber stimulation and/or restraint stress. CPT was intrathecally administered 30 min before stress exposure. Quantitative analysis of the number of c-FOS$^+$ (**F**) and pERK$^+$ (**H**) cells in superficial laminae of the SDH in each group (*n* = 4–5 mice per group; one-way ANOVA with post hoc Tukey's multiple comparisons test; *p < 0.05, ***p < 0.001, ****p < 0.0001). (**I**) Schematic illustration of the mechanisms of stress-induced mechanical pain facilitation highlighting NA signals from LC$^{\rightarrow SDH}$-NAergic terminals to *Hes5*$^+$ astrocytes and *Slc32a1*$^+$ INs. Data represent mean ± SEM. Some figure elements were created with BioRender.com.

The online version of this article includes the following source data for figure 4:

**Source data 1.** Raw numerical values for *Figure 4* plots.

signaling, which is pronociceptive, by identifying *Hes5*$^+$ astrocytes in the SDH (*Kohro et al., 2020*). Mechanistically, spinal NA acts on α$_{1A}$Rs expressed on *Hes5*$^+$ astrocytes, increases intracellular Ca$^{2+}$ levels via a G$_q$/PLC/IP$_3$ pathway, and causes mechanical hypersensitivity via release of D-serine (*Kohro et al., 2020*), an activator of NMDA receptors. This study showed that the optogenetic activation of LC$^{\rightarrow SDH}$-NA neurons produced mechanical hypersensitivity via α$_{1A}$Rs in SDH-*Hes5*$^+$ astrocytes, indicating the pronociceptive effect of LC$^{\rightarrow SDH}$-NAergic signaling. In contrast, it has also been reported that a similar opto- or chemogenetic activation produces an antinociceptive effect on thermal stimuli (*Hickey et al., 2014*; *Hirschberg et al., 2017*). The reason for these differing behavioral outcomes remains unclear; however, the stimulus modality used for the behavioral assay (mechanical vs. thermal) may be responsible, since the SDH circuits for mechanical and thermal information processing have been shown to be distinct (*Koch et al., 2018*; *Duan et al., 2014*; *Petitjean et al., 2015*; *Wang et al., 2022a*; *Warwick et al., 2022*). Alternatively, the degree of LC$^{\rightarrow SDH}$-NA neuronal activity may also be involved. Although direct comparisons between studies reporting pro- and anti-nociceptive effects are difficult, our previous studies demonstrated that intrathecal administration of high doses of NA in naive mice does not induce mechanical pain hypersensitivity and produces an antinociceptive effect on noxious heat (*Kohro et al., 2020*; *Uchiyama et al., 2022*). Indeed, a continuous stimulation of descending LC-NA neurons over several minutes has been reported to produce antinociceptive effect (*Kucharczyk et al., 2022*). Collectively, these findings indicate that the role of LC$^{\rightarrow SDH}$-NA neurons in modulating nociceptive signaling in the SDH is more complex than previously appreciated, and these differences may be related to LC$^{\rightarrow SDH}$-NA neuron activity, NA levels in the SDH, and the differential responses of SDH neurons in the superficial versus deeper laminae.

Given that SDH-INs also express α$_{1A}$Rs (*Häring et al., 2018*; *Shiraishi et al., 2021*), it is unclear why pain hypersensitivity is the preferentially manifested behavioral phenotype after activation of LC$^{\rightarrow SDH}$-NAergic signal. This study showed that NA-stimulated *Hes5*$^+$ astrocytes negatively regulate the activity of SDH-*Slc32a1*$^+$ INs. This astrocytic attenuation of SDH-*Slc32a1*$^+$ IN activity may involve adenosine signaling because A$_1$R cKD in SDH-*Slc32a1*$^+$ INs canceled the suppressive effect of A$_1$R agonist on NA-induced enhancement of *Slc32a1*$^+$-IN activity and NA-induced mechanical hypersensitivity. The adenosine-mediated inhibition of SDH-INs is consistent with the data from a previous study (*Nam et al., 2016*). In addition, we also found that knockdown of A$_1$Rs in *Slc32a1*$^+$ neurons tended to increase the proportion of *Slc32a1*$^+$ neurons with NA-induced depolarizing responses (*Figure 3—figure supplement 2*). Thus, these results suggest that spinal NA acts on α$_{1A}$Rs expressed by both *Hes5*$^+$ astrocytes and *Slc32a1*$^+$ INs; however, NA-stimulated *Hes5*$^+$ astrocytes suppress *Slc32a1*$^+$ IN activity by adenosine signaling via A$_1$Rs; this induces a state where the *Hes5*$^+$ astrocyte-derived pain-facilitating effect predominates, causing mechanical pain hypersensitivity (*Figure 4I*).

While a neuronal model for gating peripherally derived sensory signals in the SDH has been previously established by identifying several subpopulations of INs (*Todd, 2010*; *Peirs and Seal, 2016*; *Koch et al., 2018*; *Moehring et al., 2018*; *Peirs et al., 2020*), our study proposes a new regulatory mechanism in this model in which *Hes5*$^+$ astrocytes act as non-neuronal gating cells in the SDH for brain-derived NAergic signaling. This astrocytic gating control of INs may affect processing and transmission of somatosensory information in the SDH. Indeed, a pharmacological blockade of the adenosine signal to SDH-*Slc32a1*$^+$ INs suppressed the Aβ fiber-mediated c-FOS and pERK expression in the superficial SDH of stressed mice. Preproenkephalin$^+$ IN population may be among the SDH-*Slc32a1*$^+$ INs involved in stress-induced pain modulation (*François et al., 2017*). Interfering with GABA/glycine signaling in the SDH results in abnormal synaptic inputs from primary afferent Aβ fibers onto superficial SDH neurons (*Torsney and MacDermott, 2006*). Our view is also supported

by previous findings from in vivo patch-clamp recordings, which show that, in about a half of SG neurons tested, NA enhances excitatory synaptic responses evoked by tactile stimuli (air puff to the skin) (*Sonohata et al., 2019*). $\alpha_{1A}$R expression has been reported to be detected in DRG neurons, but its level seems to be much lower than in the spinal cord (*Usoskin et al., 2015*), and primary afferent glutamatergic synaptic transmission has been shown to be unaffected by $\alpha_{1A}$R agonists (*Kawasaki et al., 2003*; *Li and Eisenach, 2001*). Additionally, our hypothesis that *Hes5*[+] astrocytes co-release D-serine and ATP/adenosine in response to $\alpha_{1A}$R activation by NA is supported by previous findings in the neocortex (*Pankratov and Lalo, 2015*). Aβ fiber inputs into neurokinin 1 receptor-positive projection neurons in SDH lamina I following inhibition of GABA/glycine signaling are canceled by a pharmacological blockade of NMDA receptors (*Torsney and MacDermott, 2006*), supporting our data that the stress-induced mechanical hypersensitivity was suppressed by the antagonist of D-serine site of NMDA receptors DCK. However, we cannot exclude the possibility that D-serine also acts on NMDA receptors expressed by *Slc32a1*[+] INs. Nevertheless, given that intrathecal injection of D-serine in naive mice induces mechanical pain hypersensitivity (*Kohro et al., 2020*), it appears that the pronociceptive effect of D-serine in the SDH is primarily associated with enhanced pain processing and transmission, presumably via NMDA receptors on excitatory neurons. In addition, the effect of astrocytic adenosine on spinal pain processing has been controversial: both pronociceptive (consistent with our findings) (*Nam et al., 2016*) and antinociceptive (*Xu et al., 2021*). Such different effects might be related in part to the regionally restricted astrocytic subpopulations and factors for their activation. Pro- and anti-nociceptive effects have been reported to be mediated by astrocytes located in superficial and deeper laminae and stimulated by NA (via G-protein-coupled $\alpha_{1A}$Rs) (*Kohro et al., 2020*) and ATP (via ionotropic P2X7 receptors) (*Xu et al., 2021*), respectively. In particular, the regional differences in the role of astrocytes have been demonstrated by our previous study in which a selective chemogenetic stimulation of astrocytes in deeper SDH did not cause mechanical hypersensitivity (*Kohro et al., 2020*). Thus, it is conceivable that locally released adenosine from NA-stimulated *Hes5*[+] astrocytes, following acute restraint stress, may suppress SDH-*Slc32a1*[+] IN function, which results in mechanical pain hypersensitivity. A highly localized adenosine release from *Hes5*[+] astrocytes and the spatially restricted neuron–astrocyte communication via adenosine signaling need to be investigated in future studies.

In this study, we showed mechanical hypersensitivity by acute exposure to restraint stress and investigated its mechanisms; however, there are also many studies reporting acute stress-induced antinociception (*Supplementary file 1*). A precise explanation for the bidirectional outcomes remains elusive. However, it is widely recognized that the extent and direction of pain modulation may be due to several factors, including the nature, duration, and intensity of the stressor, as well as the sensory modality assessed in behavioral tests (*Butler and Finn, 2009*; *Jennings et al., 2014*). In fact, most studies reporting antinociceptive effects of acute restraint stress assessed behavioral responses to heat stimuli (*Supplementary file 1*). This observation is consistent with the findings in our previous study (*Uchiyama et al., 2022*) and the present study (*Figure 1—figure supplement 2*). In contrast to the robust and consistent antinociceptive effects observed with thermal stimuli, some studies evaluating behavioral responses to mechanical stimuli have reported stress-induced hypersensitivity (*Sousa et al., 2018*; *Supplementary file 1*), which aligns with our current findings. Nevertheless, acute and short-term exposure (≤1 hr) to restraint stress induces mechanical pain hypersensitivity, which was not observed with longer exposure (2 hr) (*Figure 1—figure supplement 1*) (although a 2-hr restraint stress exposure suppresses behavioral response to nociceptive thermal stimuli; *Uchiyama et al., 2022*). Thus, descending LC[→SDH]-NA neurons, *Hes5*[+] SDH astrocytes, or their output signals involved in modulating mechanical pain behavior may be attenuated by unidentified inhibitory signals evoked by longer exposure to restraint stress. A similar bidirectional behavioral outcome on mechanical pain has also been observed with intrathecal NA; low doses of NA induce mechanical pain hypersensitivity while higher doses do not (*Kohro et al., 2020*). Thus, the bidirectional change may depend on the level of NA neuronal activity and NA content in the SDH, which is supported by a previous report (*Oe et al., 2020*).

Clinically, stressor-evoked psychological and physical responses are considered critical for the development of chronic pain (*Abdallah and Geha, 2017*), especially nociplastic pain, a condition that arises from altered nociceptive signaling without actual or threatened tissue damage, or disease or lesion of the somatosensory system (*Fitzcharles et al., 2021*). In fact, patients with chronic pain

have a high incidence of stress-related psychiatric disorders, such as anxiety, fear, and depression (*Suárez-Pereira et al., 2022*; *Lerman et al., 2015*); stressful life events are known to be a key factor in establishing fibromyalgia and enhance pain in patients with irritable bowel syndrome and headaches (*Fitzcharles et al., 2021*). Thus, our findings provide a mechanistic insight into the link between stress and pain modulation and may help in understanding and developing a new strategy for chronic pain, including nociplastic pain.

There are several limitations to this study. First, under our experimental conditions, where AAV vectors were injected into both the LC and SDH (L4) to express genes in LC$^{\rightarrow SDH}$ neurons, only a small proportion (4.4%) of all LC-NA neurons could be manipulated (*Figure 1I–K*; *Figure 1—figure supplement 4A, B*), although we confirmed specific expression of DTRs in NA neurons within the LC, but not in A5 or A7 (*Figure 1—figure supplement 4C*). This low proportion is likely due to the selective targeting of LC-NA neurons that project to the L4 segment of the SDH. The total number of SDH-projecting LC-NA neurons across all spinal segments may be much higher. Given that primary afferent sensory fibers from the plantar skin of the hindpaw predominantly terminate in the L4 segment of the SDH, it is plausible that ablation of even a relatively small number of NA neurons in the LC can have a significant impact on behavior. Second, in our model (*Figure 4I*), we proposed that NA released by LC$^{\rightarrow SDH}$ NA neurons inhibits SDH-INs. Although we demonstrated that optogenetic stimulation of LC$^{\rightarrow SDH}$ NA terminals in the SDH induces endogenous NA release, we were unable to determine whether the released NA modulates the function of $Hes5^+$ astrocytes and $Slc32a1^+$ INs, or their interactions. Addressing this question would require the use of *Slc32a1-Cre* and/or *Hes5-CreERT2* mice; however, employing these lines precludes the use of *Slc6a2-Cre* mice, which are essential for specific and efficient expression of ChrimsonR in LC$^{\rightarrow SDH}$ NA neurons. Third, in our in vivo optogenetic experiments showing that stimulation of LC$^{\rightarrow SDH}$-NAergic terminals in the SDH causes mechanical pain hypersensitivity (*Figure 1N*), light could reach only the superficial laminae, even though ChrimsonR expression in LC$^{\rightarrow SDH}$-NAergic terminals was also observed in both superficial and deeper laminae of the SDH (*Figure 1—figure supplement 6*). *Kucharczyk et al., 2022* have reported that NA released from descending pathways reduces activity of wide-dynamic-range neurons in deeper laminae. Therefore, the roles of NA signaling in superficial versus deeper laminae should also be further investigated.

An important future subject is to clarify the discrepancy for the pro- and antinociceptive role of LC$^{\rightarrow SDH}$-NAergic signaling in the SDH. Since one possible candidate could be related to the sensory modality used for behavioral testing, other assays (e.g., Randall–Selitto test, CatWalk gait analysis, and weight-bearing test) should be tested. Furthermore, a difference between the duration of behavioral responses and Ca$^{2+}$ events after acute restraint stress should also be addressed in further investigation. The behavioral effect of stress on PWT (*Figure 1E*) persisted for 120 min after stress exposure, whereas Ca$^{2+}$ events changes were only observed during stress exposure (*Figure 1B*). A similar temporal difference is also observed following intraplantar injection of capsaicin (*Kohro et al., 2020*); while LC$^{\rightarrow SDH}$-NA neuron-mediated astrocytic Ca$^{2+}$ responses in SDH astrocytes last for 5–10 min after injection, behavioral hypersensitivity peaks around 60 min post-injection and gradually returns to baseline over the subsequent 60–120 min. These findings raise the possibility that astrocyte-mediated mechanical pain hypersensitivity in the SDH may involve a sustained alteration in spinal neural function, such as central sensitization.

## Methods
### Animals
Male C57BL/6J mice (CLEA Japan), male and female *Slc6a2-Cre* mice [Tg(*Slc6a2-cre*)#Stl] (kindly provided by Prof. Thomas McHugh, RIKEN Center for Brain Science) (*Wagatsuma et al., 2018*), *Hes5-CreERT2* mice [Tg(*Hes5-cre/ERT2*)2Vtlr] (kindly provided by Prof. Verdon Taylor) (*Lugert et al., 2012*), *Adra1a*flox/flox mice (*Kohro et al., 2020*), *Slc32a1-Ires-Cre* mice [*Slc32a1*tm2(cre)Lowl/J] (Stock No. 016962, The Jackson Laboratory) (*Vong et al., 2011*), and *Rosa26-LSL-tdTomato* mice [B6.Cg-*Gt(ROSA)26Sor*tm14(CAG-tdTomato)Hze/J] (Stock No. 007914, The Jackson Laboratory) (*Madisen et al., 2010*) were used. To generate *Hes5-CreERT2;Adra1a*flox/flox mice (*Kohro et al., 2020*) or *Slc32a1-Cre;Adra1a*flox/flox mice (*Uchiyama et al., 2022*), *Adra1a*flox/flox mice were crossed with *Hes5-CreERT2* mice or *Slc32a1-Ires-Cre* (*Slc32a1-Cre*) mice, and the obtained *Hes5-CreERT2;Adra1a*flox/+ mice or *Slc32a1-Cre;Adra1a*flox/+ mice were further crossed with *Adra1a*flox/flox mice. Cre-negative *Adra1a*flox/flox

littermate mice were used as controls. For induction of CreERT2 recombinase activity, *Hes5-CreERT2* mice were given an intraperitoneal (i.p.) injection of tamoxifen (#T5648, Sigma-Aldrich; 2 mg dissolved in 100 µl corn oil [#032-17016; Wako, Saitama, Japan]) once a day for 5–10 successive days. We used tamoxifen-injected mice for further analyses 7 days or more after the last injection. All mice used were 8–12 weeks of age at the start of each experiment and were housed at temperature and humidity ranges of 21–23°C and 40–60%, respectively, with a 12-hr light–dark cycle. All animals were fed food and water ad libitum. All animals were housed in standard polycarbonate cages in groups of same-sex littermates. All animal experiments were conducted according to relevant national and international guidelines contained in the 'Act on Welfare and Management of Animals' (Ministry of Environment of Japan) and 'Regulation of Laboratory Animals' (Kyushu University) and under the protocols approved by the Institutional Animal Care and Use committee review panels at Kyushu University.

### Restraint stress model

According to the methods described in our previous study (*Uchiyama et al., 2022*), mice were restrained by placing them in a Falcon 50 ml conical tube (#352070; Corning) with a hole in the tip. The space remaining behind the mouse was filled with a paper towel. Mice were allowed to breathe but not turn around. Control mice were handled briefly and placed in home cages without water or food for 1 hr.

### Measurement of behavioral response to mechanical stimuli using von Frey filaments

Mice were placed individually in an opaque plastic cylinder, which was placed on a wire mesh, and habituated for 0.5–1 hr to allow acclimatization to the new environment. After that, calibrated von Frey filaments (0.02–2.0 g; #NC12775; North Coast Medical) were applied to the plantar surface of the hindpaw of mice from below the mesh floor. 50% paw withdrawal threshold was calculated using the up-down method (*Chaplan et al., 1994*). The basal mechanical threshold was determined by performing the von Frey test before restraint stress exposure, AAV microinjection, DTX administration, optogenetic stimulation, intrathecal injection of several drugs, and tamoxifen administration. Mechanical sensitivity was evaluated at 30, 60, 90, 120, and 180 min after restraint stress or optogenetic stimulation, or at 30 min after intrathecal injection of NA or Phe.

### Measurement of behavioral response to thermal stimuli in hot-plate and paw-flick tests

For the hot-plate test (HOT/COLD PLATE ANALGESIA METER, MK-350HC, Muromachi Kikai, Tokyo, Japan), the latency to exhibit nociceptive behaviors (licking or jumping) in response to thermal stimulation (50°C) (*Uchiyama et al., 2022*; *Atwal et al., 2020*) was recorded. To prevent tissue damage, a cut-off time was set at 60 s. For the paw-flick test (Tail Flick Unit, 7360, Ugo Basile, Gemonio, Italy), the latency to elicit nociceptive responses (paw withdrawal) against heat stimulation at 40 V was recorded, with a cut-off time of 10 s to avoid tissue damage.

### Measurement of behavioral response to motor function by rotarod test

Before AAV injection, each mouse was trained on a rotarod (3 cm diameter, 8 r.p.m.; Rotarod, KN-75, Natsume Seisakusho, Tokyo, Japan) until it could remain on the apparatus for 60 s without falling (*Tsuda et al., 2009*). The mice then underwent a 60-s rotarod test both before and after DTX administration.

### Recombinant AAV (rAAV) vector production

pAAV-CAG-FLEx[GCaMP6s]-WPRE (#100842) and pAAV-synapsin (Syn)-FLEx[ChrimsonR-tdTomato]-WPRE (#62723) were purchased from Addgene. The genes encoding GCaMP6m (#40754; Addgene), Cre, GRAB$_{NE1m}$ (#123308; Addgene) and mCherry were subcloned into the pENTR plasmid (Thermo Fisher Scientific). To produce AAV vectors, we inserted GCaMP6m, GRAB$_{NE1m}$, and mCherry into pZac2.1-gfaABC$_1$D-WPRE plasmid and Cre into pZac2.1-enhanced synapsin (ESYN)-WPRE plasmid. To produce the AAV vector for the Cre-switch system, vectors containing the promoters encoding EF1α were generated from pAAV-CA-FLEx (#38042; Addgene) by substituting the promoter. We then inserted AcGFP, DTR-EGFP (kindly provided by Prof. Kenji Kohno, Nara Institute of Science and Technology), mCherry and hM3Dq (#45547; Addgene) into pAAV-EF1α-FLEx. The genes

encoding U6-sgRNA (#61593; Addgene) and SaCas9 (#78601; Addgene) were subcloned into the pENTR plasmid. Synthetic oligonucleotides including the targeting sequence for *Adora1* (5′-AAGT TCCGGGTCACCTTTCTG-3′) and non-targeting sequence (5′-CCATGTGATCGCGCTTCTCGT-3′) were replaced with the targeting site in the original pENTR-U6-sgBsa1 plasmid. The resulting U6-sgRNA cassette was transferred into pAAV-CMV-FLEx[mCherry]-WPRE plasmid to generate pAAV-CMV-FLEx[mCherry]-WPRE-U6-sgAdora1/sgYFP (sgControl). The SaCas9 cassette was transferred into pAAV-CMV-FLEx-WPRE plasmid to generate pAAV-CMV-FLEx[SaCas9]-WPRE. rAAV vectors were produced from human embryonic kidney 293T (HEK293T) cells (#632273, Takara Bio, Shiga, Japan; mycoplasma testing was not independently performed in this study; however, no signs of contamination were observed) with triple transfection [pZac or pAAV, cis plasmid; pAAV2/5 (University of Pennsylvania Gene Therapy Program Vector Core), pAAV2/9 (University of Pennsylvania Gene Therapy Program Vector Core) or pAAV2/retro (#81070; Addgene), trans plasmid; pAd DeltaF6, adenoviral helper plasmid (University of Pennsylvania Gene Therapy Program Vector Core)] and purified by two cesium chloride density gradient purification steps. The vector was dialyzed against phosphate-buffered saline (PBS) containing 0.001% (vol/vol) Pluronic-F68 (#24040032; Thermo Fisher Scientific) using Vivaspin Turbo 15 100,000 MWCO (#VS15T41; Sartorius). The genome titer of rAAV was determined by Pico Green fluorometric reagent (#P7589; Thermo Fisher Scientific) following denaturation of the AAV particles. Vectors were stored at −80°C until use.

## Intra-SDH and intra-LC injection of rAAV vectors

Viral vector injection was performed in accordance with our previously described method (*Kohro et al., 2020*; *Kohro et al., 2015*). Mice were deeply anesthetized by subcutaneous injection of ketamine (100 mg/kg) and xylazine (10 mg/kg). For intra-SDH injection, the skin was incised at Th11–L4 vertebrae, and custom-made clamps were attached to the caudal sites of the vertebral column. Paraspinal muscles around the left side of the interspace between Th13 and L1 vertebrae were removed, and the dura mater and arachnoid membrane were carefully incised using the tip of a 30-G needle to make a small window allowing a glass microcapillary to insert directly into the SDH. The microcapillary was inserted into the unilateral SDH (around 120–150 μm in depth from the surface of the dorsal root entry zone). rAAV solutions (approximately 500 nl) were injected using a Micro4 Micro Syringe Pump Controller (World Precision Instrument). After microinjection, the glass microcapillary was removed, the skin was sutured with 5-0 silk, and the mice were kept on a heating pad until recovery. For intra-LC injection, rAAV solutions were injected (approximately 300 nl in one site) adjacent to the LC (anteroposterior (AP): −5.4 mm from the bregma; mediolateral (ML): ±0.9 mm; dorsoventral (DV): −3.2 mm from the dura). Only AAV2/9-CAG-FLEx[GCaMP6s]-WPRE was injected unilaterally, and the other AAV vectors were injected bilaterally. To keep the bregma and lambda in the same horizontal plane, tolerance was maintained at <50 μm in the dorsoventral axis between the bregma/lambda. The microcapillary was withdrawn 3 min after ending the injection. The skin was then sutured with 5-0 silk, and the mice were kept on a heating pad until recovery. We used virus-injected mice for further analyses 3 weeks or more after the last injection. The following viral titers were used: AAV2/9-CAG-FLEx[GCaMP6s]-WPRE, AAV2/9-gfaABC$_1$D-GCaMP6m-WPRE, AAV2/9-EF1α-FLEx[AcGFP]-WPRE (for *Slc6a2-Cre* mice), AAV2/9-EF1α-FLEx[DTR-EGFP]-WPRE (for *Slc6a2-Cre* mice), AAV2/9-EF1α-FLEx[mCherry], AAV2/9-hSyn-FLEx[ChrimsonR-tdTomato], AAV2/9-gfaABC$_1$D-GRAB$_{NE1m}$-WPRE, AAV2/5-EF1α-FLEx[hM3Dq], AAV2/9-CMV-FLEx[SaCas9]-WPRE and AAV2/5-gfaABC$_1$D-mCherry: $1.0 \times 10^{12}$ genome copies (GC)/ml; AAV2/retro-ESYN-Cre-WPRE: $3.0 \times 10^{12}$ GC/ml; AAV2/9-EF1α-FLEx[AcGFP]-WPRE (for WT mice) and AAV2/9-EF1α-FLEx[DTR-EGFP]-WPRE (for WT mice): $2.0 \times 10^{12}$ GC/ml; AAV2/9-CMV-FLEx[mCherry]-WPRE-U6-sgYFP and AAV2/9-CMV-FLEx[mCherry]-WPRE-U6-sgAdora1: $0.5 \times 10^{12}$ GC/ml.

## Immunohistochemistry

Mice were deeply anesthetized with an i.p. injection of pentobarbital and transcardially perfused with PBS (#041-20211; Wako, Saitama, Japan) followed by ice-cold 4% paraformaldehyde (PFA; #162-16065; Wako, Saitama, Japan)/PBS. The transverse L4 segments of the spinal cord and brain were removed, postfixed in the same fixative for 3 hr (spinal cord) or overnight (brain) at 4°C, and placed in 30% sucrose solution for 24–48 hr at 4°C. After incubation, the tissues were embedded in OCT compound (#4583; Sakura Finetek Japan, Osaka, Japan) and stored at −25°C before use. Transverse

spinal cord and brain sections (30 μm) were incubated in blocking solution (3% normal goat serum [#S-1000; Vector Laboratories] or normal donkey serum [#017-000-121; Jackson ImmunoResearch]) for 2 hr at room temperature and then incubated for 48 hr at 4°C with primary antibodies: polyclonal rabbit anti-tyrosine hydroxylase (TH; 1:1000; #AB152; Millipore); polyclonal sheep anti-TH (1:1000; #AB1542; Millipore); monoclonal mouse anti-noradrenaline transporter (NET; 1:2000; #NET05-2; Mab Technologies); polyclonal rabbit anti-green fluorescent protein (GFP; 1:1000; #598; MBL International); monoclonal rabbit anti-hemagglutinin (HA)-tag (1:1000; #3724; Cell Signaling); monoclonal rat anti-glial fibrillary acidic protein (GFAP; 1:2000; #13-0300; Invitrogen); polyclonal goat anti-SRY-related high-mobility group box 9 (SOX9; 1:1000; # AF3075; R&D Systems); polyclonal goat anti-paired box 2 (PAX2; 1:500; # AF3364; R&D Systems); monoclonal rat anti-mCherry (1:2000; #M11217; Thermo Fisher Scientific); polyclonal rabbit anti-phospho-p44/42 MAPK (ERK1/2) (Thr202/Tyr204) (pERK; 1:500; #9101; Cell Signaling); anti-isolectin B4 (IB4)-biotin conjugate (1:1000; # I21414; Thermo Fisher Scientific); and monoclonal rabbit anti-c-FOS (1:5000; #226 008; Synaptic systems). After incubation, tissue sections were washed and incubated for 3 hr at room temperature with secondary antibodies (Alexa Fluor 488, 546 and/or 555; #A11001, A11008, A11035, A11056, A11081, S32351; Thermo Fisher Scientific: #ab150178; Abcam: DyLight 405; #705-475-147; Jackson ImmunoResearch). Then, tissue sections were washed, slide mounted, and subsequently placed under coverslips with VECTA-SHIELD Hardmount (Vector Laboratories). Immunofluorescence images were obtained with confocal laser microscopy (#LSM700 or LSM900; Carl Zeiss).

## Fiber photometry

Immediately after microinjection of AAV vectors encoding GCaMPs, the mice were implanted with an optic cannula. Cannulae were purchased from RWD (#R-FOC-BL400C-50NA, ø1.25 mm ferrule, ø400 μm optic fiber, 0.50 NA). The length of the optic fiber protruding from the tip of the ferrule was 4.0 mm. Optic cannula was implanted in the unilateral LC (AP: −5.4 mm from the bregma; ML: 0.9 mm; DV: −3.0 mm from the dura). Next, the cannula was secured using two types of dental glues (Super-bond; Sun Medical, Shiga, Japan: UNIFAST III; GC, Tokyo, Japan). After implantation, the skin was sutured with 5-0 silk, and the mice were kept on a heating pad until recovery. We used cannula-implanted mice for fiber photometry 3 weeks or more after the surgery and monitored $Ca^{2+}$ events in LC-NA neurons during restraint stress using the conical tube with a narrow slit (*Figure 1—video 1*). GCaMP fluorescent signals were obtained with the fiber photometry apparatus from Doric Lenses, which includes a fiber photometry console (FPC_V6), a 415-nm LED illumination (CLED_415), a 465-nm LED illumination (CLED_465), an LED driver (LEDD_2), and a fluorescence mini cube with built-in fluorescence photodetector amplifier (iFMC4-G2_IE(410-420)_E(460–490)_F(500–550)_S). A 415-nm light was modulated at 208 Hz, while a 465-nm light was modulated at 572 Hz. The optic cannula implanted in mice and fluorescence mini cube were connected by a low-autofluorescence patch cable (#MAF1L1; Thorlabs). The power output at the fiber tip was 30–40 μW. After connecting to the patch cable, the mice were allowed to move freely in the home cage for at least 10 min for habituation. After habituation, fluorescence recording was started. Emitted signals from the tissue were collected through the same fiber and sampled at 12 kHz.

## Processing and analysis of fiber photometry data

To avoid the effects of photobleaching, data from the first 5 min of recording were discarded. The raw data were then decimated to 30 Hz (down-sampled by 400) and low-pass Butterworth-filtered at 2 Hz. The 415 nm excitation channel served as an isosbestic, $Ca^{2+}$-independent control wavelength for GCaMP, allowing for bleaching and movement artifact corrections when directly fitted to the $Ca^{2+}$-dependent 465 nm channel. The Correction Baseline function from Doric Neuroscience Studio (Doric Lenses) was used to fit the 415 nm signal to the 465 nm signal using an adaptive iterative re-weighted Penalized Least Squares algorithm (*Zhang et al., 2010*). The fluorescence change was expressed as a relative change, $\Delta F/F = (F_{465} - F_{415})/F_{415}$, where $F_{465}$ is the 465 nm-induced $Ca^{2+}$-dependent signal, and $F_{415}$ is the 415 nm-induced $Ca^{2+}$-independent signal. During restraint stress, the $Ca^{2+}$ transients and escape behaviors (struggling) occurred almost simultaneously (*Figure 1—video 1*), but the 415 nm-induced $Ca^{2+}$-independent signals were unchanged. Robust $Z$-scores were then calculated as $Z = (\Delta F/F - \text{Median } \Delta F/F)/\text{NIQR}$, where Median $\Delta F/F$ is the median of $\Delta F/F$ during the entire measurement period, and NIQR is the normalized interquartile range (0.7413 × interquartile range). $Ca^{2+}$

events were detected with a threshold of $Z$-score >3. The frequency of $Ca^{2+}$ events was calculated as the number of $Ca^{2+}$ events per minute in the 10 min before stress, 60 min during stress (10 min × 6 times), and 10 min after stress.

## Drug administration

To ablate DTR$^+$ neurons, DTX (10 µg/kg in PBS; #048-34371; Wako, Saitama, Japan) was administered i.p. for two successive days 3 weeks after intra-LC injection of AAV2/9-EF1α-FLEx[DTR-EGFP] or [AcGFP]-WPRE. We used DTX-injected mice for further analyses 2 weeks or more after the last injection. For intrathecal injection, a 30-G needle attached to a Hamilton microsyringe was inserted between the L5/L6 vertebrae and then punctured through the dura. The following drugs and doses were used for intrathecal injection experiments: silodosin (3 nmol in 5 µl PBS; #191-17591; Wako, Saitama, Japan; injection was given 30 min before optogenetic stimulation); L-norepinephrine hydrochloride (0.1 nmol in 5 µl saline; #74480; Sigma-Aldrich); 5,7-dichlorokynurenic acid (DCK; 10 nmol in 5 µl saline of 2% dimethylsulfoxide (DMSO); #0286; Tocris; injection was given immediately before the start of exposure to restraint stress); (R)-(−)-phenylephrine hydrochloride (Phe; 0.05 nmol in 5 µl saline; #163-11791; Wako, Saitama, Japan); CPT (3 nmol in 5 µl saline of 2% DMSO; #C102; Sigma-Aldrich; injection was given immediately before the start of exposure to restraint stress or 30 min before intrathecal injection of NA).

## In vivo optogenetic stimulation

Immediately after microinjection of AAV vectors encoding ChrimsonR, mice were implanted with an optic cannula. A cannula comprised a ceramic ferrule (#CFLC230-10; Thorlabs; ø1.25 mm, 6.4 mm length) and an optic fiber (#FT200UMT; Thorlabs; ø200 µm, 0.39 NA). The length of the optic fiber protruding from the tip of the ferrule was less than 0.3 mm. The vertebrae were immobilized in place using a metal bar along the Th12–L1. A small laminectomy was performed through the Th13 bone using a dental drill. The cannula was secured above the dura mater using two types of dental glues (Super-bond; Sun Medical: UNIFAST III; GC). In this procedure, the dura mater and spinal cord parenchyma were left intact and unscathed. After implantation, the skin was sutured with 5-0 silk, and the mice were kept on a heating pad until recovery. We used cannula-implanted mice for optogenetic stimulation 3 weeks or more after the surgery. To activate ChrimsonR in LC$^{\rightarrow SDH}$-NA axons/terminals, 625 nm LED light (#M625F2; Thorlabs) was delivered through a ferrule patch cable (#M83L01; Thorlabs). Light stimulations (2 mW, 10 Hz, 5 ms pulse duration, 5 s light on, 15 s light off, 10 cycles) were generated by an LED driver (#LEDD_2; Doric Lenses).

## Fluorescent $Ca^{2+}$ and NA imaging in spinal cord slices

Fluorescent imaging was performed in accordance with our previously described method (*Kohro et al., 2020*). GCaMP6m- and GRAB$_{NE1m}$-expressing mice were anesthetized with an i.p. injection of urethane (1.2–1.5 mg/kg), and lumbosacral laminectomy was performed. The spinal cord (L1–S2) was removed and placed in a cold, high-sucrose, artificial cerebrospinal (aCSF) fluid (27 mM NaHCO$_3$, 1.4 mM NaH$_2$PO$_4$, 2.5 mM KCl, 7.0 mM MgSO$_4$, 1.0 mM CaCl$_2$, 222 mM sucrose, and 0.5 mM ascorbic acid), which was bubbled with 95% O$_2$ and 5% CO$_2$. After cutting all ventral and dorsal roots, transverse L4 spinal cord slices (350 µm thick) were made using a vibrating microtome (#VT1200; Leica); the slices were kept in oxygenated aCSF solution (125 mM NaCl, 1.25 mM NaH$_2$PO$_4$, 2.5 mM KCl, 1.0 mM MgCl$_2$, 2.0 mM CaCl$_2$, 26 mM NaHCO$_3$, and 20 mM glucose) at room temperature (22–25°C) for at least 30 min before use. The slices were placed in a recording chamber, and fluorescent signals were measured using a confocal laser scanning microscope (#FV3000; Olympus, Tokyo, Japan). The chamber was perfused with aCSF saturated with 95% O$_2$ and 5% CO$_2$ at 31–32°C and at 2–3 ml/min using peristaltic pumps. A 488 nm diode laser was used for the excitation of GCaMP6m and GRAB$_{NE1m}$. The size of the image was 509 × 509 µm$^2$ (512 × 512 pixels). The recordings were acquired at a rate of 1.083 s per image. Light stimulations (1 mW, 10 Hz, 5 ms pulse duration, 1–20 s) were generated by an LED driver (#LEDD_2; Doric Lenses) and delivered through an optic fiber (Thorlabs, FT200UMT, ø200 µm, 0.39 NA). For an experiment on $\alpha_{1A}R$ inhibition, optogenetic stimulations (10 s) were performed twice (at least 15 min apart) in the same slice. Silodosin (40 nM in aCSF) was continuously applied by bath application starting 5 min before the onset of 2nd optogenetic stimulation. After optogenetic stimulation, NA (1 or 10 µM in aCSF) was applied by bath application as the positive control.

## Processing and analysis of Ca²⁺ and NA imaging data

Videos were imported into ImageJ Fiji (https://imagej.net/software/fiji/), and motion artifacts were corrected using TurboReg (*Thévenaz et al., 1998*). The fluorescence change was expressed as a relative percentage change, $\Delta F/F = 100 \times (F_t - F_0)/F_0$, where $F_t$ is the fluorescence at time $t$, and $F_0$ is the baseline average for 30 frames before the start of stimulation. For Ca²⁺ imaging using GCaMP, the size of each region of interest (ROI) was set to a circle with a diameter of 10 µm (*Kohro et al., 2020*). ROIs were manually identified based on the following criteria: located within 200 µm from the surface of gray matter and ≥30% $\Delta F/F$ fluorescence change after optogenetic stimulation or NA application. For the experiment on $\alpha_{1A}R$ inhibition, normalized $\Delta F/F$ was calculated as ($\Delta F/F$ in 2nd stimulation)/($\Delta F/F$ in 1st stimulation). For NA imaging using GRAB$_{NE1m}$, the raw image was binarized, and a single ROI was detected from such a binarized image with the following parameters: contiguous pixel number >100 and 0 < circularity < 1 (*Oe et al., 2020*). Calculation of $\Delta F/F$ in ROIs was performed using the Time Series Analyzer V3 plugin in ImageJ.

## Whole-cell patch-clamp recordings using spinal cord slices

According to our previously described method (*Uchiyama et al., 2022*), mice were deeply anesthetized with urethane (1.2–1.5 mg/kg, i.p.), and the lumbar spinal cord was removed and placed in a cold high-sucrose aCSF (250 mM sucrose, 2.5 mM KCl, 2 mM CaCl₂, 2 mM MgCl₂, 1.2 mM NaH₂PO₄, 25 mM NaHCO₃, and 11 mM glucose). Parasagittal and transverse spinal cord slices (250–300 µm thick) were made with a vibrating microtome (VT1200, Leica) and then the slices kept in oxygenated aCSF solution (125 mM NaCl, 2.5 mM KCl, 2 mM CaCl₂, 1 mM MgCl₂, 1.25 mM NaH₂PO₄, 26 mM NaHCO₃, and 20 mM glucose) at room temperature (22–25°C) for at least 30 min. The spinal cord slice was then put into a recording chamber, where it was continuously superfused with aCSF solution at 25–28°C at a flow rate of 4–6 ml/min. Recordings were made with the Axopatch 700B amplifier and pCLAMP 10.4 acquisition software (Molecular Devices). Data were digitized with an analog-to-digital converter (Digidata 1550; Molecular Devices), stored on a personal computer with a data acquisition program (ClampeX version 10.4; Molecular Devices), and analyzed with a software package (Clampfit version 10.7; Molecular Devices). sIPSCs were recorded in the voltage-clamp mode at a holding potential of 0 mV. Patch pipettes were filled with an internal solution (120 mM CsMeSO₄, 15 mM CsCl, 10 mM HEPES, 5 mM QX-314, 4 mM MgATP, 0.3 mM Na₂GTP, 0.2 mM EGTA, 10 mM TEA-Cl and 8 mM NaCl [pH 7.28] adjusted with CsOH), and whole-cell patch-clamp recordings were made from SG neurons. Resting membrane potentials (RMPs) were recorded in current-clamp mode. Patch pipettes were filled with an internal solution (125 mM K-gluconate, 10 mM KCl, 0.5 mM EGTA, 10 mM HEPES, 4 mM ATP-Mg, 0.3 mM NaGTP, 10 mM phosphocreatine, pH 7.28 adjusted with KOH), and whole-cell patch-clamp recordings were made from tdTomato⁺ or mCherry⁺ neurons. The following drugs were used: NA (20 µM; #74480; Sigma-Aldrich), clozapine-*N*-oxide (CNO; 100 µM; #BML-NS105; Enzo Life Sciences), CPT (1 µM), N6-cyclopentyladenosine (CPA; 1 µM; #119135; Sigma-Aldrich), CNQX disodium salt hydrate (CNQX; 10 µM; #C239; Sigma-Aldrich), (+)-MK-801 hydrogen maleate (MK-801; 20 µM; #M107; Sigma-Aldrich), 1(S), 9(R)-(−)-bicuculline methiodide (10 µM; #14340; Sigma-Aldrich) and strychnine (1 µM; #S0532; Sigma-Aldrich). All drugs were dissolved in aCSF solution. CNO was superfused for 3 min. The bath application of CPT was continuously superfused from 4 min before CNO application. NA was superfused with an antagonist cocktail (CNQX, MK-801, bicuculline, and strychnine) for 3–5 min to block synaptic inputs from other neurons to the recorded *Slc32a1*⁺ neurons (*Wu et al., 2004*; *Liu et al., 2010*). If the recording neuron was depolarized by NA perfusion, then CPA was co-perfused with NA and an antagonist cocktail for 3 min. The frequency of sIPSCs for 1 min of pre- and post-NA application was quantified using Clampfit version 10.7 (Molecular Devices). We quantified averaged RMP for 1 min of pre- and post-drug application, and a change in RMP (ΔRMP) of 5 mV or more was judged to be depolarization or hyperpolarization.

## In situ hybridization

Mice were deeply anesthetized with an i.p. injection of pentobarbital and transcardially perfused with PBS followed by ice-cold 4% PFA/PBS. The L4 spinal cord was removed and postfixed in the same fixative 24 hr at 4°C. After that, tissues were incubated with 10%, 20%, and 30% sucrose solutions at 4°C, embedded in OCT compound, and stored at −25°C before use. The L4 segments were sectioned at a thickness of 14 µm. In situ hybridization was performed using RNAscope Multiplex Fluorescent

Reagent Kit v2 (#323100; ACDbio), according to the manufacturer's protocol for fixed frozen tissue. The following probes were used: Mm-*Adora1* (#402261; ACDbio); Mm-*Slc32a1*-C3 (#319191-C3; ACDbio). Tissue sections were analyzed using an LSM700 Imaging System. Cells were considered positive if three or more punctate dots were present in the nucleus and/or cytoplasm (*Shiraishi et al., 2021*).

## Electrical stimulation of Aβ fibers and counting of c-FOS⁺ and pERK⁺ neurons in the SDH

For pERK immunofluorescence, but not for c-FOS, after exposure to 1-hr restraint stress, mice were deeply anesthetized with an i.p. injection of urethane (1.2–1.5 mg/kg). Thirty min after stress, Aβ fiber stimulation was performed. The electrodes were attached to the plantar and dorsal surfaces of the left hindpaw. Transcutaneous nerve stimuli were applied using a stimulus generator (#STG4002-16 mA; Multi Channel Systems). The current intensity of the 2000 Hz stimuli was 1000 μA (*Matsumoto et al., 2008*). Mice were fixed at 2 (for pERK) or 90 min (for c-FOS) after electrical stimulation. For c-FOS immunostaining, mice were deeply anesthetized with an i.p. injection of pentobarbital prior to fixation. For quantification of c-FOS⁺ and pERK⁺ cells, three sections from the L4 spinal cord segments were randomly selected from each mouse, and the number of c-FOS⁺ and pERK⁺ neurons in the superficial laminae I–IIi (defined by staining of IB4) was counted. Consistent with a previous study (*Matsumoto et al., 2008*), we confirmed that electrical stimulation of Aδ and C fibers, but not Aβ fibers, induced ERK phosphorylation in SDH neurons (data not shown).

## Statistical analysis

Statistical analyses were performed using Prism (GraphPad). Quantitative data were expressed as the mean ± SEM. Statistical significance of differences was determined by using two-tailed paired *t*-test (*Figure 1—figure supplement 1*, *Figure 1—figure supplement 2E*, and *Figure 1—figure supplement 5*), Mann–Whitney test (*Figure 2F*, *Figure 1—figure supplement 4B*), Wilcoxon signed-rank test (*Figure 3L, N*, *Figure 3—figure supplement 1C*), Fisher's exact test (*Figure 3—figure supplement 2*), Friedman test with Dunn's multiple comparisons test (*Figures 1E and 2E*), one-way ANOVA with Dunnett's multiple comparisons test (*Figure 2C*), one-way ANOVA with Tukey's multiple comparisons test (*Figure 4F, H*) and repeated measures two-way ANOVA with Bonferroni's multiple comparisons test (*Figures 1B, H, K, N*, *2G–J*, *3A–C*, *4A–D*, *Figure 1—figure supplements 2E and 5*), as appropriate, after determining the normality (Kolmogorov–Smirnov test or Shapiro–Wilk test). p values are indicated as \*p < 0.05, \*\*p < 0.01, \*\*\*p < 0.001, \*\*\*\*p < 0.0001.

## Acknowledgements

We would like to thank Editage for editing a draft of this manuscript, and the University of Pennsylvania vector core for providing pZac2.1, pAAV2/5, pAAV2/9, and pAd DeltaF6 plasmid. This work was supported by Japan Society for the Promotion of Science (JSPS) KAKENHI Grants JP19H05658 (MT), JP20H05900 (MT), and JP24H00067 (MT), by the Core Research for Evolutional Science and Technology (CREST) program from AMED under Grant Number 25gm1510013h (MT), and by Research Support Project for Life Science and Drug Discovery (Basis for Supporting Innovative Drug Discovery and Life Science Research (BINDS)) from AMED under Grant Number JP25ama121031 (MT). RK-K, SU, and KY were JSPS research fellows (JP22KJ2471, JP23KJ1729, and JP19J21063, respectively).

## Additional information

### Funding

| Funder | Grant reference number | Author |
| --- | --- | --- |
| Japan Society for the Promotion of Science | JP19H05658 | Makoto Tsuda |
| Japan Society for the Promotion of Science | JP20H05900 | Makoto Tsuda |

| Funder | Grant reference number | Author |
|---|---|---|
| Japan Society for the Promotion of Science | JP24H00067 | Makoto Tsuda |
| Japan Society for the Promotion of Science | JP22KJ2471 | Riku Kawanabe-Kobayashi |
| Japan Society for the Promotion of Science | JP23KJ1729 | Sawako Uchiyama |
| Japan Society for the Promotion of Science | JP19J21063 | Kohei Yoshihara |
| Japan Agency for Medical Research and Development | 25gm1510013h | Makoto Tsuda |
| Japan Agency for Medical Research and Development | JP25ama121031 | Makoto Tsuda |

The funders had no role in study design, data collection, and interpretation, or the decision to submit the work for publication.

### Author contributions

Riku Kawanabe-Kobayashi, Sawako Uchiyama, Formal analysis, Funding acquisition, Investigation, Visualization, Methodology, Writing – original draft, Writing – review and editing; Kohei Yoshihara, Formal analysis, Funding acquisition, Investigation, Methodology, Writing – original draft; Keisuke Koga, Funding acquisition, Investigation, Methodology; Daiki Kojima, Investigation; Thomas J McHugh, Izuho Hatada, Resources; Ko Matsui, Kenji F Tanaka, Methodology; Makoto Tsuda, Conceptualization, Formal analysis, Supervision, Visualization, Writing – original draft, Project administration, Writing – review and editing

### Author ORCIDs

Sawako Uchiyama ⓘ https://orcid.org/0009-0002-7074-4788
Thomas J McHugh ⓘ https://orcid.org/0000-0002-1243-5189
Kenji F Tanaka ⓘ https://orcid.org/0000-0003-2511-0057
Makoto Tsuda ⓘ https://orcid.org/0000-0003-0585-9570

### Ethics

All animal experiments were conducted according to relevant national and international guidelines contained in the 'Act on Welfare and Management of Animals' (Ministry of Environment of Japan) and 'Regulation of Laboratory Animals' (Kyushu University) and under the protocols approved by the Institutional Animal Care and Use committee review panels at Kyushu University (A23-206-0, A23-208-1, A23-330-1, A24-064-0, and A24-461-0).

Reviewer #1 (Public review): https://doi.org/10.7554/eLife.104453.3.sa1
Reviewer #2 (Public review): https://doi.org/10.7554/eLife.104453.3.sa2
Reviewer #3 (Public review): https://doi.org/10.7554/eLife.104453.3.sa3
Author response https://doi.org/10.7554/eLife.104453.3.sa4

## Additional files

### Supplementary files

Supplementary file 1. Summary of anti- and pro-nociceptive effects of acute restraint stress reported in previously published studies. Previously reported changes in pain-related behaviors induced by acute restraint stress were summarized along with information on animal species, strain, sex, duration of restraint, and pain tests.

MDAR checklist

### Data availability

All data associated with this study are present in the paper or the Supplemental Information.

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
