## [Editor Report · eLife Assessment]

This **important** study identifies a novel role for Hes5+ astrocytes in modulating the activity of descending pain-inhibitory noradrenergic neurons from the locus coeruleus during stress-induced pain facilitation. The role of glia in modulating neurological circuits including pain is poorly understood, and in that light, the role of Hes5+ astrocytes in this circuit is a key finding with broader potential impacts. This work is supported by **convincing** evidence, albeit somewhat limited by the indirect nature of the evidence linking adenosine to nearby neuronal modulation, and possible questions on the population specificity of the transgenic approach.

---

## [Referee Report · Reviewer #1 (Public review)]

Review of the revised submission:

I thank the authors for their detailed consideration of my comments and for the additional data, analyses, and clarifications they have incorporated. The new behavioral experiments, quantification of targeted manipulations, and expanded methodological details strengthen the manuscript and address many of my initial concerns. While some questions remain for future work, the authors' careful responses and the additional evidence provided help resolve the main issues I raised, and I am generally satisfied with the revisions.

Review of original submission:

Summary

In this article, Kawanabe-Kobayashi et al., aim to examine the mechanisms by which stress can modulate pain in mice. They focus on the contribution of noradrenergic neurons (NA) of the locus coeruleus (LC). The authors use acute restraint stress as a stress paradigm and found that following one hour of restraint stress mice display mechanical hypersensitivity. They show that restraint stress causes the activation of LC NA neurons and the release of NA in the spinal cord dorsal horn (SDH). They then examine the spinal mechanisms by which LC→SDH NA produces mechanical hypersensitivity. The authors provide evidence that NA can act on alphaA1Rs expressed by a class of astrocytes defined by the expression of Hes (Hes+). Furthermore, they found that NA, presumably through astrocytic release of ATP following NA action on alphaA1Rs Hes+ astrocytes, can cause an adenosine-mediated inhibition of SDH inhibitory interneurons. They propose that this disinhibition mechanism could explain how restraint stress can cause the mechanical hypersensitivity they measured in their behavioral experiments.

Strengths:

(1) Significance. Stress profoundly influences pain perception; resolving the mechanisms by which stress alters nociception in rodents may explain the well-known phenomenon of stress-induced analgesia and/or facilitate the development of therapies to mitigate the negative consequences of chronic stress on chronic pain.

(2) Novelty. The authors' findings reveal a crucial contribution of Hes+ spinal astrocytes in the modulation of pain thresholds during stress.

(3) Techniques. This study combines multiple approaches to dissect circuit, cellular, and molecular mechanisms including optical recordings of neural and astrocytic Ca2+ activity in behaving mice, intersectional genetic strategies, cell ablation, optogenetics, chemogenetics, CRISPR-based gene knockdown, slice electrophysiology, and behavior.

Weaknesses:

(1) Mouse model of stress. Although chronic stress can increase sensitivity to somatosensory stimuli and contribute to hyperalgesia and anhedonia, particularly in the context of chronic pain states, acute stress is well known to produce analgesia in humans and rodents. The experimental design used by the authors consists of a single one-hour session of restraint stress followed by 30 min to one hour of habituation and measurement of cutaneous mechanical sensitivity with von Frey filaments. This acute stress behavioral paradigm corresponds to the conditions in which the clinical phenomenon of stress-induced analgesia is observed in humans, as well as in animal models. Surprisingly, however, the authors measured that this acute stressor produced hypersensitivity rather than antinociception. This discrepancy is significant and requires further investigation.

(2) Specifically, is the hypersensitivity to mechanical stimulation also observed in response to heat or cold on a hotplate or coldplate?

(3) Using other stress models, such as a forced swim, do the authors also observe acute stress-induced hypersensitivity instead of stress-induced antinociception?

(4) Measurement of stress hormones in blood would provide an objective measure of the stress of the animals.

(5) Results:

(a) Optical recordings of Ca2+ activity in behaving rodents are particularly useful to investigate the relationship between Ca2+ dynamics and the behaviors displayed by rodents.

(b) The authors report an increase in Ca2+ events in LC NA neurons during restraint stress: Did mice display specific behaviors at the time these Ca2+ events were observed such as movements to escape or orofacial behaviors including head movements or whisking?

(c) Additionally, are similar increases in Ca2+ events in LC NA neurons observed during other stressful behavioral paradigms versus non-stressful paradigms?

(d) Neuronal ablation to reveal the function of a cell population.

(e) The proportion of LC NA neurons and LC→SDH NA neurons expressing DTR-GFP and ablated should be quantified (Figures 1G and J) to validate the methods and permit interpretation of the behavioral data (Figures 1H and K). Importantly, the nocifensive responses and behavior of these mice in other pain assays in the absence of stress (e.g., hotplate) and a few standard assays (open field, rotarod, elevated plus maze) would help determine the consequences of cell ablation on processing of nociceptive information and general behavior.

(f) Confirmation of LC NA neuron function with other methods that alter neuronal excitability or neurotransmission instead of destroying the circuit investigated, such as chemogenetics or chemogenetics, would greatly strengthen the findings. Optogenetics is used in Figure 1M, N but excitation of LC→SDH NA neuron terminals is tested instead of inhibition (to mimic ablation), and in naïve mice instead of stressed mice.

(g) Alpha1Ars. The authors noted that "Adra1a mRNA is also expressed in INs in the SDH".

(h) The authors should comprehensively indicate what other cell types present in the spinal cord and neurons projecting to the spinal cord express alpha1Ars and what is the relative expression level of alpha1Ars in these different cell types.

(i) The conditional KO of alpha1Ars specifically in Hes5+ astrocytes and not in other cell types expressing alpha1Ars should be quantified and validated (Figure 2H).

(j) Depolarization of SDH inhibitory interneurons by NA (Figure 3). The authors' bath applied NA, which presumably activates all NA receptors present in the preparation.

k) The authors' model (Figure 4H) implies that NA released by LC→SDH NA neurons leads to the inhibition of SDH inhibitory interneurons by NA. In other experiments (Figure 1L, Figure 2A), the authors used optogenetics to promote the release of endogenous NA in SDH by LC→SDH NA neurons. This approach would investigate the function of NA endogenously released by LC NA neurons at presynaptic terminals in the SDH and at physiological concentrations and would test the model more convincingly compared to the bath application of NA.

(l) As for other experiments, the proportion of Hes+ astrocytes that express hM3Dq, and the absence of expression in other cells, should be quantified and validated to interpret behavioral data.

(m) Showing that the effect of CNO is dose-dependent would strengthen the authors' findings.

(n) The proportion of SG neurons for which CNO bath application resulted in a reduction in recorded sIPSCs is not clear.

(o) A1Rs. The specific expression of Cas9 and guide RNAs, and the specific KD of A1Rs, in inhibitory interneurons but not in other cell types expressing A1Rs should be quantified and validated.

(6) Methods:

It is unclear how fiber photometry is performed using "optic cannula" during restraint stress while mice are in a 50ml falcon tube (as shown in Figure 1A).

---

## [Referee Report · Reviewer #2 (Public review)]

Summary:

This study investigates the role of spinal astrocytes in mediating stress-induced pain hypersensitivity, focusing on the LC (locus coeruleus)-to-SDH (spinal dorsal horn) circuit and its mechanisms. The authors aimed to delineate how LC activity contributes to spinal astrocytic activation under stress conditions, explore the role of noradrenaline (NA) signaling in this process, and identify the downstream astrocytic mechanisms that influence pain hypersensitivity.

The authors provide strong evidence that 1-hour restraint stress-induced pain hypersensitivity involves the LC-to-SDH circuit, where NA triggers astrocytic calcium activity via alpha1a adrenoceptors (alpha1aRs). Blockade of alpha1aRs on astrocytes-but not on Vgat-positive SDH neurons-reduced stress-induced pain hypersensitivity. These findings are rigorously supported by well-established behavioral models and advanced genetic techniques, uncovering the critical role of spinal astrocytes in modulating stress-induced pain.

However, the study's third aim-to establish a pathway from astrocyte alpha1aRs to adenosine-mediated inhibition of SDH-Vgat neurons-is less compelling. While pharmacological and behavioral evidence is intriguing, the ex vivo findings are indirect and lack a clear connection to the stress-induced pain model. Despite these limitations, the study advances our understanding of astrocyte-neuron interactions in stress-pain contexts and provides a strong foundation for future research into glial mechanisms in pain hypersensitivity.

Strengths:

The study is built on a robust experimental design using a validated 1-hour restraint stress model, providing a reliable framework to investigate stress-induced pain hypersensitivity. The authors utilized advanced genetic tools, including retrograde AAVs, optogenetics, chemogenetics, and subpopulation-specific knockouts, allowing precise manipulation and interrogation of the LC-SDH circuit and astrocytic roles in pain modulation. Clear evidence demonstrates that NA triggers astrocytic calcium activity via alpha1aRs, and blocking these receptors effectively reduces stress-induced pain hypersensitivity.

Weaknesses:

The study offers mainly indirect evidence for astrocyte-released adenosine acting on SDH-VGAT neurons. The potential contributions of astrocyte-derived D-serine and adenosine to different spinal neuron subtypes, as well as the transient "dip" in astrocytic calcium following LC optostimulation, merit further clarification in future work once appropriate tools become available.

Comments on revisions:

The authors have thoroughly addressed my previous comments, resolving most of the points I raised except those noted in the "Weaknesses" section above. I understand that some of these aspects will require future tool development.

---

## [Referee Report · Reviewer #3 (Public review)]

Summary

This is an exciting and timely study addressing the role of descending noradrenergic systems in nocifensive responses. While it is well-established that spinally released noradrenaline (aka norepinephrine) generally acts as an inhibitory factor in spinal sensory processing, this system is highly complex. Descending projections from the A6 (locus coeruleus, LC) and the A5 regions typically modulate spinal sensory processing and reduce pain behaviours, but certain subpopulations of LC neurons have been shown to mediate pronociceptive effects, such as those projecting to the prefrontal cortex (Hirshberg et al., PMID: 29027903).

The study proposes that descending cerulean noradrenergic neurons potentiate touch sensation via alpha-1 adrenoceptors on Hes5+ spinal astrocytes, contributing to mechanical hyperalgesia. This finding is consistent with prior work from the same group (dd et al., PMID:). However, caution is needed when generalising about LC projections, as the locus coeruleus is functionally diverse, with differences in targets, neurotransmitter co-release, and behavioural effects. Specifying the subpopulations of LC neurons involved would significantly enhance the impact and interpretability of the findings.

Strengths

The study employs state-of-the-art molecular, genetic, and neurophysiological methods, including precise CRISPR and optogenetic targeting, to investigate the role of Hes5+ astrocytes. This approach is elegant and highlights the often-overlooked contribution of astrocytes in spinal sensory gating. The data convincingly support the role of Hes5+ astrocytes as regulators of touch sensation, coordinated by brain-derived noradrenaline in the spinal dorsal horn, opening new avenues for research into pain and touch modulation.

Furthermore, the data support a model in which superficial dorsal horn (SDH) Hes5+ astrocytes act as non-neuronal gating cells for brain-derived noradrenergic (NA) signalling through their interaction with substantia gelatinosa inhibitory interneurons. Locally released adenosine from NA-stimulated Hes5+ astrocytes, following acute restraint stress, may suppress the function of SDH-Vgat+ inhibitory interneurons, resulting in mechanical pain hypersensitivity. However, the spatially restricted neuron-astrocyte communication underlying this mechanism requires further investigation in future studies.

Comments on revisions:

One important point remains insufficiently resolved. In Figure S4C, two of the three visible neurons in the A5 example appear to show a white "halo" at the cell border, suggesting a merge of eGFP (green) and TH (magenta) and therefore possible transgene positivity. To draw a confident conclusion about the specificity of the approach for the A6 (LC) population, the authors are kindly asked to provide high-resolution images of several representative A5 sections, presented both as merged and as separate colour channels. Ideally, quantification across multiple rostrocaudal sections of A5, A6 and A7 should be provided. This is essential for determining whether any transgene expression occurs within the A5 nucleus, particularly given its several-millimetre rostrocaudal extent. As the behavioural phenotype arises from manipulation of only a small subset of A6 neurons, ruling out any contribution from A5 (or A7) is critical for validating pathway specificity, especially in light of prior reports showing that similar approaches can label A5 fibres.

---

## [Author Response]

The following is the authors’ response to the original reviews.

**Public reviews:**

**Reviewer #1 (Public review):**
Summary:In this article, Kawanabe-Kobayashi et al., aim to examine the mechanisms by which stress can modulate pain in mice. They focus on the contribution of noradrenergic neurons (NA) of the locus coeruleus (LC). The authors use acute restraint stress as a stress paradigm and found that following one hour of restraint stress mice display mechanical hypersensitivity. They show that restraint stress causes the activation of LC NA neurons and the release of NA in the spinal cord dorsal horn (SDH). They then examine the spinal mechanisms by which LC→SDH NA produces mechanical hypersensitivity. The authors provide evidence that NA can act on alphaA1Rs expressed by a class of astrocytes defined by the expression of Hes (Hes+). Furthermore, they found that NA, presumably through astrocytic release of ATP following NA action on alphaA1Rs Hes+ astrocytes, can cause an adenosine-mediated inhibition of SDH inhibitory interneurons. They propose that this disinhibition mechanism could explain how restraint stress can cause the mechanical hypersensitivity they measured in their behavioral experiments.Strengths:(1) Significance. Stress profoundly influences pain perception; resolving the mechanisms by which stress alters nociception in rodents may explain the well-known phenomenon of stress-induced analgesia and/or facilitate the development of therapies to mitigate the negative consequences of chronic stress on chronic pain.(2) Novelty. The authors' findings reveal a crucial contribution of Hes+ spinal astrocytes in the modulation of pain thresholds during stress.(3) Techniques. This study combines multiple approaches to dissect circuit, cellular, and molecular mechanisms including optical recordings of neural and astrocytic Ca2+ activity in behaving mice, intersectional genetic strategies, cell ablation, optogenetics, chemogenetics, CRISPR-based gene knockdown, slice electrophysiology, and behavior.Weaknesses:(1) Mouse model of stress. Although chronic stress can increase sensitivity to somatosensory stimuli and contribute to hyperalgesia and anhedonia, particularly in the context of chronic pain states, acute stress is well known to produce analgesia in humans and rodents. The experimental design used by the authors consists of a single one-hour session of restraint stress followed by 30 min to one hour of habituation and measurement of cutaneous mechanical sensitivity with von Frey filaments. This acute stress behavioral paradigm corresponds to the conditions in which the clinical phenomenon of stress-induced analgesia is observed in humans, as well as in animal models. Surprisingly, however, the authors measured that this acute stressor produced hypersensitivity rather than antinociception. This discrepancy is significant and requires further investigation.

We thank the reviewer for evaluating our work and for highlighting both its strengths and weaknesses. As stated by the reviewer, numerous studies have reported acute stress-induced antinociception. However, as shown in a new additional table (Table S1) in which we have summarized previously published data using the acute restraint stress model employed in our present study, most studies reporting antinociceptive effects of acute restraint stress assessed behavioral responses to heat stimuli or formalin. This observation is consistent with the findings from our previous study (Uchiyama et al., Mol Brain, 2022 (PMID: 34980215)). The present study also confirms that acute restraint stress reduces behavioral responses to noxious heat (see also our response to Comment #2 below). In contrast to the robust and consistent antinociceptive effects observed with thermal stimuli, some studies evaluating behavioral responses to mechanical stimuli have reported stress-induced hypersensitivity (see Table S1), which aligns with our current findings. Taken together, these data support our original notion that the effects of acute stress on pain-related behaviors depend on several factors, including the nature, duration, and intensity of the stressor, as well as the sensory modality assessed in behavioral tests. We have incorporated this discussion and Table S1 into the revised manuscript (lines 344-353). Furthermore, we have slightly modified the text including the title, replacing "pain facilitation" with "mechanical pain hypersensitivity" to more accurately reflect our research focus and the conclusion of this study that LC^→SDH^ NAergic signaling to spinal astrocytes is required for stress-induced mechanical pain hypersensitivity. Finally, while mouse models of stress could provide valuable insights, the clinical relevance of stress-induced mechanical pain hypersensitivity remains to be elucidated and requires further investigation. We hope these clarifications address your concerns.

(2) Specifically, is the hypersensitivity to mechanical stimulation also observed in response to heat or cold on a hotplate or coldplate?

Thank you for your important comment. We have now conducted additional behavioral experiments to assess responses to heat using the hot-plate test. We found that mice subjected to restraint stress did not exhibit behavioral hypersensitivity to heat stimuli; instead, they displayed antinociceptive responses (Figure S2; lines 95-98). These results are consistent with our previous findings (Uchiyama et al., Mol Brain, 2022 (PMID: 34980215)) as well as numerous other reports (Table S1).

(3) Using other stress models, such as a forced swim, do the authors also observe acute stress-induced hypersensitivity instead of stress-induced antinociception?

As suggested by the reviewer, we conducted a forced swim test. We found that mice subjected to forced swimming, which has been reported to produce analgesic effects on thermal stimuli (Contet et al., Neuropsychopharmacology, 2006 (PMID: 16237385)), did not exhibit any changes in mechanical pain hypersensitivity (Figure S2; lines 98-99). Furthermore, a previous study demonstrated that mechanical pain sensitivity is enhanced by other stress models, such as exposure to an elevated open platform for 30 min (Kawabata et al., Neuroscience, 2023 (PMID: 37211084)). However, considering our data showing that changes in mechanosensory behavior induced by restraint stress depend on the duration of exposure (Figure S1), and that restraint stress also produced an antinociceptive effect on heat stimuli (Figure S2), stress-induced modulation of pain is a complex phenomenon influenced by multiple factors, including the stress model, intensity, and duration, as well as the sensory modality used for behavioral testing (lines 100-103).

(4) Measurement of stress hormones in blood would provide an objective measure of the stress of the animals.

A previous study has demonstrated that plasma corticosterone levels—a stress hormone—are elevated following a 1-hour exposure to restraint stress in mice (Kim et al., Sci Rep, 2018 (PMID: 30104581)), using a stress protocol similar to that employed in our current study. We have included this information with citing this paper (lines 104-105).

(5) Results:(a) Optical recordings of Ca2+ activity in behaving rodents are particularly useful to investigate the relationship between Ca2+ dynamics and the behaviors displayed by rodents.

In the optical recordings of Ca^2+^ activity in LC neurons, we monitored mouse behavior during stress exposure. We have now included a video of this in the revised manuscript (video; lines 111-114).

(b) The authors report an increase in Ca2+ events in LC NA neurons during restraint stress: Did mice display specific behaviors at the time these Ca2+ events were observed such as movements to escape or orofacial behaviors including head movements or whisking?

By reanalyzing the temporal relationship between Ca^2+^ events and mouse behavior during stress exposure, we found that the Ca^2+^ transients and escape behaviors (struggling) occurred almost simultaneously (video). A similar temporal correlation is also observed in Ca^2+^ responses in the bed nucleus of the stria terminalis (Luchsinger et al., Nat Commun, 2021 (PMID: 34117229)). The video file has been included in the revised manuscript (video; lines 111-113, 552-553, 573-575).

Additionally, as described in the Methods section and shown in Figure S2 of the initial version (now Figure S3), non-specific signals or artifacts—such as those caused by head movements—were corrected (although such responses were minimal in our recordings).

(c) Additionally, are similar increases in Ca2+ events in LC NA neurons observed during other stressful behavioral paradigms versus non-stressful paradigms?

We appreciate the reviewer's valuable suggestion. Since the present, initial version of our manuscript focused on acute restraint stress, we did not measure Ca^2+^ events in LC-NA neurons in other stress models, but a recent study has shown an increase in Ca^2+^ responses in LC-NA neurons by social defeat stress (Seiriki et al., BioRxiv, https://www.biorxiv.org/content/10.1101/2025.03.07.641347v1).

(d) Neuronal ablation to reveal the function of a cell population.

This method has been widely used in numerous previous studies as an effective experimental approach to investigate the role of specific neuronal populations—including SDH-projecting LC-NA neurons (Ma et al., Brain Res, 2022 (PMID: 34929182); Kawanabe et al., Mol Brain, 2021 (PMID: 33971918))—in CNS function.

(e) The proportion of LC NA neurons and LC→SDH NA neurons expressing DTR-GFP and ablated should be quantified (Figures 1G and J) to validate the methods and permit interpretation of the behavioral data (Figures 1H and K). Importantly, the nocifensive responses and behavior of these mice in other pain assays in the absence of stress (e.g., hotplate) and a few standard assays (open field, rotarod, elevated plus maze) would help determine the consequences of cell ablation on processing of nociceptive information and general behavior.

As suggested, we conducted additional experiments to quantitatively analyze the number of LC^→SDH^-NA neurons. We used WT mice injected with AAVretro-Cre into the SDH (L4 segment) and AAV-FLEx[DTR-EGFP] into the LC. In these mice, 4.4% of total LC-NA neurons [positive for tyrosine hydroxylase (TH)] expressed DTR-GFP, representing the LC^→SDH^-NA neuronal population (Figure S4; lines 126-127). Furthermore, treatment with DTX successfully ablated the DTR-expressing LC^→SDH^-NA neurons. Importantly, the neurons quantified in this analysis were specifically those projecting to the L4 segment of the SDH; therefore, the total number of SDH-projecting LC-NA neurons across all spinal segments is expected to be much higher.

We also performed the rotarod and paw-flick tests to assess motor function and thermal sensitivity following ablation of LC^→SDH^-NA neurons. No significant differences were observed between the ablated and control groups (Figure S5; lines 131-134), indicating that ablation of these neurons does not produce non-specific behavioral deficits in motor function or other sensory modalities.

(f) Confirmation of LC NA neuron function with other methods that alter neuronal excitability or neurotransmission instead of destroying the circuit investigated, such as chemogenetics or chemogenetics, would greatly strengthen the findings. Optogenetics is used in Figure 1M, N but excitation of LCLC^→SDH^ NA neuron terminals is tested instead of inhibition (to mimic ablation), and in naïve mice instead of stressed mice.

We appreciate the reviewer’s comment. The optogenetic approach is useful for manipulating neuronal excitability; however, prolonged light illumination (> tens of seconds) can lead to undesirable tissue heating, ionic imbalance, and rebound spikes (Wiegert et al., Neuron, 2017 (PMID: 28772120)), making it difficult to apply in our experiments, in which mice are exposed to stress for 60 min. For this reason, we decided to employ the cell-ablation approach in stress experiments, as it is more suitable than optogenetic inhibition. In addition, as described in our response to weakness (1)-a by Reviewer 3 (Public review), we have now demonstrated the specific expression of DTRs in NA neurons in the LC, but not in A5 or A7 (Figure S4; lines 127-128), confirming the specificity of LCLC^→SDH^-NAergic pathway targeting in our study. Chemogenetics represent another promising approach to further strengthen our findings on the role of LCLC^→SDH^-NA neurons, but this will be an important subject for future studies, as it will require extensive experiments to assess, for example, the effectiveness of chemogenetic inhibition of these neurons during 60 min of restraint stress, as well as optimization of key parameters (e.g., systemic DCZ doses).

(g) Alpha1Ars. The authors noted that "Adra1a mRNA is also expressed in INs in the SDH".

The expression of α_1A_Rs in inhibitory interneurons in the SDH is consistent with our previous findings (Uchiyama et al., Mol Brain, 2022 (PMID: 34980215)) as well as with scRNA-seq data (http://linnarssonlab.org/dorsalhorn/, Häring et al., Nat Neurosci, 2018 (PMID: 29686262)).

(h) The authors should comprehensively indicate what other cell types present in the spinal cord and neurons projecting to the spinal cord express alpha1Ars and what is the relative expression level of alpha1Ars in these different cell types.

According to the scRNA-seq data (https://seqseek.ninds.nih.gov/genes, Russ et al., Nat Commun, 2021 (PMID: 34588430); http://linnarssonlab.org/dorsalhorn/, Häring et al., Nat Neurosci, 2018 (PMID: 29686262)), we confirmed that α_1A_Rs are predominantly expressed in astrocytes and inhibitory interneurons in the spinal cord. Also, an α_1A_R-expressing excitatory neuron population (Glut14) expresses *Tacr1*, *GPR83*, and *Tac1* mRNAs, markers that are known to be enriched in projection neurons of the SDH. This raises the possibility that α_1A_ Rs may also be expressed in a subset of projection neurons, although further experiments are required to confirm this. In DRG neurons, α_1A_R expression was detected to some extent, but its level seems to be much lower than in the spinal cord (http://linnarssonlab.org/drg/ Usoskin et al., Nat Neurosci, 2015 (PMID: 25420068)). Consistent with this, primary afferent glutamatergic synaptic transmission has been shown to be unaffected by α_1A_R agonists (Kawasaki et al., Anesthesiology, 2003 (PMID: 12606912); Li and Eisenach, JPET, 2001 (PMID: 11714880)). This information has been incorporated into the Discussion section (lines 317-319).

(i) The conditional KO of alpha1Ars specifically in Hes5+ astrocytes and not in other cell types expressing alpha1Ars should be quantified and validated (Figure 2H).

We have previously shown a selective KO of α_1A_R in Hes5^+^ astrocytes in the same mouse line (Kohro et al., Nat Neurosci, 2020 (PMID: 33020652)). This information has been included in the revised text (line 166-167).

(j) Depolarization of SDH inhibitory interneurons by NA (Figure 3). The authors' bath applied NA, which presumably activates all NA receptors present in the preparation.

We believe that the reviewer’s concern may pertain to the possibility that NA acts on non-Vgat^+^ neurons, thereby indirectly causing depolarization of Vgat^+^ neurons. As described in the Method section of the initial version, in our electrophysiological experiments, we added four antagonists for excitatory and inhibitory neurotransmitter receptors—CNQX (AMPA receptor), MK-801 (NMDA receptor), bicuculline (GABA_A_ receptor), and strychnine (glycine receptor)—to the artificial cerebrospinal fluid to block synaptic inputs from other neurons to the recorded Vgat^+^ neurons. Since this method is widely used for this purpose in many previous studies (Wu et al., J Neurosci, 2004 (PMID: 15140934); Liu et al., Nat Neurosci, 2010 (PMID: 20835251)), it is reasonable to conclude that NA directly acts on the recorded SDH Vgat^+^ interneurons to produce excitation (lines 193-196).

(k) The authors' model (Figure 4H) implies that NA released by LC→SDH NA neurons leads to the inhibition of SDH inhibitory interneurons by NA. In other experiments (Figure 1L, Figure 2A), the authors used optogenetics to promote the release of endogenous NA in SDH by LC→SDH NA neurons. This approach would investigate the function of NA endogenously released by LC NA neurons at presynaptic terminals in the SDH and at physiological concentrations and would test the model more convincingly compared to the bath application of NA.

We appreciate the reviewer’s valuable comment. As noted, optogenetic stimulation of LC^→SDH^-NA neurons would indeed be useful to test this model. However, in our case, it is technically difficult to investigate the responses of Vgat^+^ inhibitory neurons and Hes5^+^ astrocytes to NA endogenously released from LC^→SDH^-NA neurons. This would require the use of *Vgat-Cre* or *Hes5-CreERT2* mice, but employing these lines precludes the use of NET-Cre mice, which are necessary for specific and efficient expression of ChrimsonR in LC^→SDH^-NA neurons. Nevertheless, all of our experimental data consistently support the proposed model, and we believe that the reviewer will agree with this, without additional experiments that is difficult to conduct because of technical limitations (lines 382-388).

(l) As for other experiments, the proportion of Hes+ astrocytes that express hM3Dq, and the absence of expression in other cells, should be quantified and validated to interpret behavioral data.

We thank the reviewer for raising this point. In our experiments, we used an HA-tag (fused with hM3Dq) to confirm hM3Dq expression. However, it is difficult to precisely analyze individual astrocytes because, as shown in Figure 3J, the boundaries of many HA-tag^+^ astrocytes are indistinguishable. This seems to be due to the membrane localization of HA-tag, the complex morphology of astrocytes, and their tile-like distribution pattern (Baldwin et al., Trends Cell Biol, 2024 (PMID: 38180380)). Nevertheless, our previous study demonstrated that ~90% of astrocytes in the superficial laminae are Hes5^+^ (Kohro et al., Nat Neurosci, 2020 (PMID: 33020652)), and intra-SDH injection of AAV-hM3Dq labeled the majority of superficial astrocytes (Figure 3J). Thus, AAV-FLEx[hM3Dq] injection into *Hes5-CreERT2* mice allows efficient expression of hM3Dq in Hes5^+^ astrocytes in the SDH. Importantly, our previous studies using *Hes5-CreERT2* mice have confirmed that hM3Dq is not expressed in other cell types (neurons, oligodendrocytes, or microglia) (Kohro et al., Nat Neurosci, 2020 (PMID: 33020652); Kagiyama et al., Mol Brain, 2025 (PMID: 40289116)). This information regarding the cell-type specificity has now been briefly described in the revised version (lines 218-219).

(m) Showing that the effect of CNO is dose-dependent would strengthen the authors' findings.

Thank you for your comment. We have now demonstrated a dose-dependent effect of CNO on Ca^2+^ responses in SDH astrocytes (please see our response to Major Point (4) from Reviewer #2 (Recommendations for the Authors) (Figure S7; lines 225-228)). In addition, we also confirmed that the effect of CNO is not nonspecific, as CNO application did not alter sIPSCs in spinal cord slices prepared from mice lacking hM3Dq expression in astrocytes (Figure S7; lines 225-228).

(n) The proportion of SG neurons for which CNO bath application resulted in a reduction in recorded sIPSCs is not clear.

We have included individual data points in each bar graph to more clearly illustrate the effect of CNO on each neuron (Figure 3L, N).

(o) A1Rs. The specific expression of Cas9 and guide RNAs, and the specific KD of A1Rs, in inhibitory interneurons but not in other cell types expressing A1Rs should be quantified and validated.

In addition to the data demonstrating the specific expression of SaCas9 and sgAdora1 in Vgat^+^ inhibitory neurons shown in Figure 3G of the initial version, we have now conducted the same experiments with a different sample and confirmed this specificity: SaCas9 (detected via HA-tag) and sgAdora1 (detected via mCherry) were expressed in PAX2^+^ inhibitory neurons (Author response image 1). Furthermore, as shown in Figure 3H and I in the initial version, the functional reduction of A_1_Rs in inhibitory neurons was validated by electrophysiological recordings. Together, these results support the successful deletion of A_1_Rs in inhibitory neurons.

**Author response image 1. sa4fig1:** Expression of HA-tag and mCherry in inhibitory neurons (a different sample from Figure 3G) SaCas9 (yellow, detected by HA-tag) and mCherry (magenta) expression in the PAX2^+^ inhibitory neurons (cyan) at 3 weeks after intra-SDH injection of AAV-FLEx[SaCas9-HA] and AAV-FLEx[mCherry]-U6-sgAdora1 in *Vgat-Cre* mice. Arrowheads indicate genome-editing Vgat^+^ cells. Scale bar, 25 µm.

(6) Methods:It is unclear how fiber photometry is performed using "optic cannula" during restraint stress while mice are in a 50ml falcon tube (as shown in Figure 1A).

We apologize for the omission of this detail in the Methods section. To monitor Ca^2+^ events in LC-NA neurons during restraint stress, we created a narrow slit on the top of the conical tube, allowing mice to undergo restraint stress while connected to the optic fiber (see video). This information has now been added to the Methods section (lines 552-553).

**Recommendations for the authors:**

**Reviewer #1 (Recommendations for the authors):**
(1) Scientific rigor:It is unclear if the normal distribution of the data was determined before selecting statistical tests.

We apologize for omitting this description. For all statistical analyses in this study, we first assessed the normality of the data and then selected appropriate statistical tests accordingly. We have added this information to the revised manuscript (lines 711-712).

(2) Nomenclature:(a) Mouse Genome Informatics (MGI) nomenclature should be used to describe mouse genotypes (i.e., gene name in italic, only first letter is capitalized, alleles in superscript).(b) FLEx should be used instead of flex.

Thank you for the suggestion. We have corrected these terms (including FLEx) according to MGI nomenclature.

**Reviewer #2 (Public review):**
Summary:This study investigates the role of spinal astrocytes in mediating stress-induced pain hypersensitivity, focusing on the LC (locus coeruleus)-to-SDH (spinal dorsal horn) circuit and its mechanisms. The authors aimed to delineate how LC activity contributes to spinal astrocytic activation under stress conditions, explore the role of noradrenaline (NA) signaling in this process, and identify the downstream astrocytic mechanisms that influence pain hypersensitivity.The authors provide strong evidence that 1-hour restraint stress-induced pain hypersensitivity involves the LC-to-SDH circuit, where NA triggers astrocytic calcium activity via alpha1a adrenoceptors (alpha1aRs). Blockade of alpha1aRs on astrocytes - but not on Vgat-positive SDH neurons - reduced stress-induced pain hypersensitivity. These findings are rigorously supported by well-established behavioral models and advanced genetic techniques, uncovering the critical role of spinal astrocytes in modulating stress-induced pain.However, the study's third aim - to establish a pathway from astrocyte alpha1aRs to adenosine-mediated inhibition of SDH-Vgat neurons - is less compelling. While pharmacological and behavioral evidence is intriguing, the ex vivo findings are indirect and lack a clear connection to the stress-induced pain model. Despite these limitations, the study advances our understanding of astrocyte-neuron interactions in stress-pain contexts and provides a strong foundation for future research into glial mechanisms in pain hypersensitivity.Strengths:The study is built on a robust experimental design using a validated 1-hour restraint stress model, providing a reliable framework to investigate stress-induced pain hypersensitivity. The authors utilized advanced genetic tools, including retrograde AAVs, optogenetics, chemogenetics, and subpopulation-specific knockouts, allowing precise manipulation and interrogation of the LC-SDH circuit and astrocytic roles in pain modulation. Clear evidence demonstrates that NA triggers astrocytic calcium activity via alpha1aRs, and blocking these receptors effectively reduces stress-induced pain hypersensitivity.Weaknesses:Despite its strengths, the study presents indirect evidence for the proposed NA-to-astrocyte(alpha1aRs)-to-adenosine-to-SDH-Vgat neurons pathway, as the link between astrocytic adenosine release and stress-induced pain remains unclear. The ex vivo experiments, including NA-induced depolarization of Vgat neurons and chemogenetic stimulation of astrocytes, are challenging to interpret in the stress context, with the high CNO concentration raising concerns about specificity. Additionally, the role of astrocyte-derived D-serine is tangential and lacks clarity regarding its effects on SDH Vgat neurons. The astrocyte calcium signal "dip" after LC optostimulation-induced elevation are presented without any interpretation.

We appreciate the reviewer's careful reading of our paper. According to the reviewer's comments, we have performed new additional experiments and added some discussion in the revised manuscript (please see the point-by-point responses below).

**Reviewer #2 (Recommendations for the authors):**
The astrocyte-mediated pathway of NA-to-astrocyte (alpha1aRs)-to-adenosine-to-SDH Vgat neurons (A1R) in the context of stress-induced pain hypersensitivity requires more direct evidence. While the data showing that the A1R agonist CPT inhibits stress-induced hypersensitivity and that stress combined with Aβ fiber stimulation increases pERK in the SDH are intriguing, these findings primarily support the involvement of A1R on Vgat neurons and are only behaviorally consistent with SDH-Vgat neuronal A1R knockdown. The role of astrocytes in this pathway in vivo remains indirect. The ex vivo chemogenetic Gq-DREADD stimulation of SDH astrocytes, which reduced sIPSCs in Vgat neurons in a CPT-dependent manner, needs revision with non-DREADD+CNO controls to validate specificity. Furthermore, the ex vivo bath application of NA causing depolarization in Vgat neurons, blocked by CPT, adds complexity to the data leaving me wondering how astrocytes are involved in such processes, and it does not directly connect to stress-induced pain hypersensitivity. These findings are potentially useful but require additional refinement to establish their relevance to the stress model.

We thank the reviewer for the insightful feedback. First, regarding the role of astrocytes in this pathway in vivo, we showed in the initial version that mechanical pain hypersensitivities induced by intrathecal NA injection and by acute restraint stress were attenuated by both pharmacological blockade and Vgat^+^ neuron-specific knockdown of A_1_Rs (Figure 4A, B). Given that NA- and stress-induced pain hypersensitivity is mediated by α_1A_R-dependent signaling in Hes5^+^ astrocytes (Kohro et al., Nat Neurosci, 2020 (PMID: 33020652); this study), these findings provide in vivo evidence supporting the involvement of the NA → Hes5^+^ astrocyte (via α_1A_Rs) → adenosine → Vgat^+^ neuron (via A_1_Rs) pathway. As noted in the reviewer’s major comment (2), in vivo monitoring of adenosine dynamics in the SDH during stress exposure would further substantiate the astrocyte-to-neuron signaling pathway. However, we did not detect clear signals, potentially due to several technical limitations (see our response below). Acknowledging this limitation, we have now added a new paragraph in the end of Discussion section to address this issue. Second, the specificity of the effect of CNO has now been validated by additional experiments (see our response to major point (4)). Third, the reviewer’s concern regarding the action of NA on Vgat^+^ neurons has also been addressed (see our response to major point (3) below).

Major points:(1) The in vivo pharmacology using DCK to antagonize D-serine signaling from alpha1a-activated astrocytes is tangential, as there is limited evidence on how Vgat neurons (among many others) respond to D-serine. This aspect requires more focused exploration to substantiate its relevance.

We propose that the site of action of D-serine in our neural circuit model is the NMDA receptors (NMDARs) on excitatory neurons, a notion supported by our previous findings (Kohro et al., Nat Neurosci, 2020 (PMID: 33020652); Kagiyama et al., Mol Brain, 2025 (PMID: 40289116)). However, we cannot exclude the possibility that D-serine also acts on NMDARs expressed by Vgat^+^ inhibitory neurons. Nevertheless, given that intrathecal injection of D-serine in naïve mice induces mechanical pain hypersensitivity (Kohro et al., Nat Neurosci, 2020 (PMID: 33020652)), it appears that the pronociceptive effect of D-serine in the SDH is primarily associated with enhanced pain processing and transmission, presumably via NMDARs on excitatory neurons. We have added this point to the Discussion section in the revised manuscript (lines 325-330).

(2) Additionally, employing GRAB-Ado sensors to monitor adenosine dynamics in SDH astrocytes during NA signaling would significantly strengthen conclusions about astrocyte-derived adenosine's role in the stress model.

We agree with the reviewer’s comment. Following this suggestion, we attempted to visualize NA-induced adenosine (and ATP) dynamics using GRAB-ATP and GRAB-Ado sensors (Wu et al., Neuron, 2022 (PMID: 34942116); Peng et al., Science, 2020 (PMID: 32883833)) in acutely isolated spinal cord slices from mice after intra-SDH injection of AAV-hSyn-GRABATP_1.0_ and -GRABAdo_1.0_. We confirmed expression of these sensors in the SDH (Author response image 2a) and observed increased signals after bath application of ATP (0.1 or 1 µM) or adenosine (1 µM) (Author response image 2b, c). However, we were unable to detect clear signals following NA stimulation (Author response image 2b, c). The reason for this lack of detectable changes remains unclear. If the release of adenosine from astrocytes is a highly localized phenomenon, it may be measurable using high-resolution microscopy capable of detecting adenosine levels at the synaptic level and more sensitive sensors. Further investigation will therefore be required (lines 340-341).

**Author response image 2. sa4fig2:** Ex vivo imaging of GRAB-ATP and GRAB-Ado sensors. (a) Representative images of GRAB_ATP1.0_ (left, green) or GRAB_Ado1.0_ (right, green) expression in the SDH at 3 weeks after SDH injection of AAV-hSyn-GRAB_Ado1.0_ or AAV-hSyn-GRAB_Ado1.0_ in *Hes5-CreERT2* mice. Scale bar, 200 µm. (b) Left: Representative fluorescence images showing GRAB_ATP1.0_ responses before and after perfusion with NA or ATP. Right: Representative traces showing responses to ATP (0.1 and 1 µM) or NA (10 µM). (c) Left: Representative fluorescence images showing GRABAdo1.0 responses before and after perfusion with NA or adenosine (Ado). Right: Representative traces showing responses to Ado (0.01, 0.1, and 1 µM), NA (10 µM), or no application (negative control).

(3) The interpretation of Figure 3D is challenging. The manuscript implies that 20 μM NA acts on Adra1a receptors on Vgat neurons to depolarize them, but this concentration should also activate Adra1a on astrocytes, leading to adenosine release and potential inhibition of depolarization. The observation of depolarization despite these opposing mechanisms requires explanation, as does the inhibition of depolarization by bath-applied A1R agonist. Of note, 20 μM NA is a high concentration for Adra1a activation, typically responsive at nanomolar levels. The discussion should reconcile this with prior studies indicating dose-dependent effects of NA on pain sensitivity (e.g., Reference 22).

Like the reviewer, we also considered that bath-applied NA could activate α_1A_Rs expressed on Hes5^+^ astrocytes. To clarify this point, we have performed additional patch-clamp recordings and found that knockdown of A_1_Rs in Vgat^+^ neurons tended to increase the proportion of Vgat^+^ neurons with NA-induced depolarizing responses (Figure S8). Therefore, it is conceivable that NA-induced excitation of Vgat^+^ neurons may involve both a direct effect of NA activating α_1A_Rs in Vgat^+^ neurons and an indirect inhibitory signaling from NA-stimulated Hes5^+^ astrocytes via adenosine (lines 298-300).

The concentration of NA used in our ex vivo experiments is higher than that typically used in vitro with αR-_1A_expressing cell lines or primary culture cells, but is comparable to concentrations used in other studies employing spinal cord slices (Kohro et al., Nat Neurosci, 2020 (PMID: 33020652); Baba et al., Anesthesiology, 2000 (PMID: 10691236); Lefton et al., Science, 2025 (PMID: 40373122)). In slice experiments, drugs must diffuse through the tissue to reach target cells, resulting in a concentration gradient. Therefore, higher drug concentrations are generally necessary in slice experiments, in contrast to cultured cell experiments, where drugs are directly applied to target cells. Importantly, we have previously shown that the pharmacological effects of 20 μM NA on Vgat^+^ neurons and Hes5^+^ astrocytes are abolished by loss of α_1A_Rs in these cells (Uchiyama et al., Mol Brain, 2022 (PMID: 34980215); Kohro et al., Nat Neurosci, 2020 (PMID: 33020652)), confirming the specificity of these NA actions.

Regarding the dose-dependent effect of NA on pain sensitivity, NA-induced pain hypersensitivity is abolished in Hes5^+^ astrocyte-specific α_1A_R-KO mice (Kohro et al., Nat Neurosci, 2020 (PMID: 33020652)), indicating that this behavior is mediated by α_1A_Rs expressed on Hes5^+^ astrocytes. In contrast, the suppression of pain sensitivity by high doses of NA was unaffected in the KO mice (Kohro et al., Nat Neurosci, 2020 (PMID: 33020652)), suggesting that other adrenergic receptors may contribute to this phenomenon. Clarifying the responsible receptors will require future investigation.

(4) In Figure 3K-M, the CNO concentration used (100 μM) is unusually high compared to standard doses (1 to a few μM), raising concerns about potential off-target effects. Including non-hM3Dq controls and using lower CNO concentrations are essential to validate the specificity of the observed effects. Similarly, the study should clarify whether astrocyte hM3Dq stimulation alone (without NA) would induce hyperpolarization in Vgat neurons and how this interacts with NA-induced depolarization.

We acknowledge that the concentration of CNO used in our experiments is relatively high compared to that used in other reports. However, in our experiments, application of CNO at 1, 10, and 100 μM induced Ca^2+^ increases in GCaMP6-expressing astrocytes in spinal cord slices in a concentration-dependent manner (Figure S7). Among these, 100 μM CNO most effectively replicated the NA-induced Ca^2+^ signals in astrocytes. Based on these findings, we selected this concentration for use in both the current and previous studies (Kohro et al., Nat Neurosci., 2020 (PMID: 33020652)). Importantly, to rule out non-specific effects, we conducted control experiments using spinal cord slices from mice that did not express hM3Dq in astrocytes and confirmed that CNO had no effect on Ca^2+^ responses in astrocytes and sIPSCs in substantial gelatinosa (SG) neurons (Figure S7; lines 223-228). Thus, although the CNO concentration used is relatively high, the observed effects of CNO are not non-specific but result from the chemogenetic activation of hM3Dq-expressing astrocytes.

In this study, we used *Hes5-CreERT2* and *Vgat-Cre* mice to manipulate gene expression in Hes5^+^ astrocytes and Vgat^+^ neurons, respectively. In order to fully address the reviewer’s comment, the use of both Cre lines is necessary. However, simultaneous and independent genetic manipulation in each cell type using Cre activity alone is not feasible with the current genetic tools. We have mentioned this as a technical limitation in the Discussion section (lines 382-388).

(5) The role of D-serine released by hM3Dq-stimulated astrocytes in (separately) modulating sub-types of neurons including excitatory neurons and Vgat positives needs more detailed discussion. If no effect of D-serine on Vgat neurons is observed, this should be explicitly stated, and the discussion should address why this might be the case.

As mentioned in our response to Major Point (1) above, we have added a discussion of this point in the revised manuscript (lines 325-330).

(6) Finally, the observed "dip" in astrocyte calcium signals below baseline following the large peaks with LC optostimulation should be discussed further, as understanding this phenomenon could provide valuable insights into astrocytic signaling dynamics in the context of single acute or repetitive chronic stress.

Thank you for your comment. We found that this phenomenon was not affected by pretreatment with the α_1A_R-specific antagonist silodosin (Author response image 3), which effectively suppressed Ca^2+^ elevations evoked by stimulation of LC-NA neurons (Figure 2F). This implies that the phenomenon is independent of α_1A_R signaling. Elucidating the detailed underlying mechanism remains an important direction for future investigation.

**Author response image 3. sa4fig3:** The observed "dip" in astrocyte Ca^2+^ signals was not affected by pretreatment with the α_1A_R-specific antagonist silodosin. Representative traces of astrocytic GCaMP6m signals in response to optogenetic stimulation of LC-NAe^→SDH^rgic axons/terminals in a spinal cord slice. Each trace shows the GCaMP6m signal before and after optogenetic stimulation (625 nm, 1 mW, 10 Hz, 5 ms pulse duration, 10 s). Slices were pretreated with silodosin (40 nM) for 5 min prior to stimulation.

**Reviewer #3 (Public review):**
Summary:This is an exciting and timely study addressing the role of descending noradrenergic systems in nocifensive responses. While it is well-established that spinally released noradrenaline (aka norepinephrine) generally acts as an inhibitory factor in spinal sensory processing, this system is highly complex. Descending projections from the A6 (locus coeruleus, LC) and the A5 regions typically modulate spinal sensory processing and reduce pain behaviours, but certain subpopulations of LC neurons have been shown to mediate pronociceptive effects, such as those projecting to the prefrontal cortex (Hirshberg et al., PMID: 29027903).The study proposes that descending cerulean noradrenergic neurons potentiate touch sensation via alpha-1 adrenoceptors on Hes5+ spinal astrocytes, contributing to mechanical hyperalgesia. This finding is consistent with prior work from the same group (dd et al., PMID:). However, caution is needed when generalising about LC projections, as the locus coeruleus is functionally diverse, with differences in targets, neurotransmitter co-release, and behavioural effects. Specifying the subpopulations of LC neurons involved would significantly enhance the impact and interpretability of the findings.Strengths:The study employs state-of-the-art molecular, genetic, and neurophysiological methods, including precise CRISPR and optogenetic targeting, to investigate the role of Hes5+ astrocytes. This approach is elegant and highlights the often-overlooked contribution of astrocytes in spinal sensory gating. The data convincingly support the role of Hes5+ astrocytes as regulators of touch sensation, coordinated by brain-derived noradrenaline in the spinal dorsal horn, opening new avenues for research into pain and touch modulation.Furthermore, the data support a model in which superficial dorsal horn (SDH) Hes5+ astrocytes act as non-neuronal gating cells for brain-derived noradrenergic (NA) signalling through their interaction with substantia gelatinosa inhibitory interneurons. Locally released adenosine from NA-stimulated Hes5+ astrocytes, following acute restraint stress, may suppress the function of SDH-Vgat+ inhibitory interneurons, resulting in mechanical pain hypersensitivity. However, the spatially restricted neuron-astrocyte communication underlying this mechanism requires further investigation in future studies.Weaknesses(1) Specificity of the LC Pathway targetingThe main concern lies with how definitively the LC pathway was targeted. Were other descending noradrenergic nuclei, such as A5 or A7, also labelled in the experiments? The authors must convincingly demonstrate that the observed effects are mediated exclusively by LC noradrenergic terminals to substantiate their claims (i.e. "we identified a circuit, the descending LC→SDH-NA neurons").(a) For instance, the direct vector injection into the LC likely results in unspecific effects due to the extreme heterogeneity of this nucleus and retrograde labelling of the A5 and A7 nuclei from the LC (i.e., Li et al., PMID: 26903420).

We appreciate the reviewer's valuable comments. To address this point, we performed additional experiments and demonstrated that intra-SDH injection of AAVretro-Cre followed by intra-LC injection of AAV2/9-EF1α-FLEx[DTR-EGFP] specifically results in DTR expression in NA neurons of the LC, but not of the A5 or A7 regions (Figure S4; lines 127-128). These results confirm the specificity of targeting the LC^→SDH^-NAergic pathway in our study.

(b) It is difficult to believe that the intersectional approach described in the study successfully targeted LC→SDH-NA neurons using AAVrg vectors. Previous studies (e.g., PMID: 34344259 or PMID: 36625030) demonstrated that similar strategies were ineffective for spinal-LC projections. The authors should provide detailed quantification of the efficiency of retrograde labelling and specificity of transgene expression in LC neurons projecting to the SDH.

Thank you for your comment. As we described in our response to the weakness (5)-e of Reviewer #1 (Public review), our additional analysis showed that, under our experimental conditions, expression of genes (for example DTR) was observed in 4.4% of NA (TH^+^) neurons in the LC (Figure S4; lines 126-127).

The reasons for this difference between the previous studies and our current study is unclear; however, it is likely attributed to methodological differences, including the type of viral vectors employed, species differences (mouse (PMID: 34344259, our study) vs. rat (PMID: 36625030)), the amount of AAV injected into the SDH (300 nL at three sites (PMID: 34344259), and 300 nL at a single site (our study)) and LC (500 nL at a single site (PMID: 34344259), and 300 nL at a single site (our study)), as well as the depth of AAV injection in the SDH (200–300 µm from the dorsal surface of the spinal cord (PMID: 34344259), and 120–150 µm in depth from the surface of the dorsal root entry zone (our study)).

(c) Furthermore, it is striking that the authors observed a comparably strong phenotypical change in Figure 1K despite fewer neurons being labelled, compared to Figure 1H and 1N with substantially more neurons being targeted. Interestingly, the effect in Figure 1K appears more pronounced but shorter-lasting than in the comparable experiment shown in Figure 1H. This discrepancy requires further explanation.

Although only a representative section of the LC was shown in the initial version, LC^→SDH^-NA neurons are distributed rostrocaudally throughout the LC, as previously reported (Llorca-Torralba et al., Brain, 2022 (PMID: 34373893)). Our additional experiments analyzing multiple sections of the anterior and posterior regions of the LC have now revealed that approximately sixty LC^→SDH^-NA neurons express DTR, and these neurons are eliminated following DTX treatment (Figure S4; lines 126-128) (it should be noted that these neurons specifically project to the L4 segment of the SDH, and the total number of LC^→SDH^-NA neurons is likely much higher). Considering the specificity of LC^→SDH^-NAergic pathway targeting demonstrated in our study (as described above), together with the fact that primary afferent sensory fibers from the plantar skin of the hindpaw predominantly project to the L4 segment of the SDH, these data suggest that the observed behavioral changes are attributable to the loss of these neurons and that ablation of even a relatively small number of NA neurons in the LC can have a significant impact on behavior. We have added this hypothesis in the Discussion section (lines 373-382).

Regarding the data in Figures 1H and 1K, as the reviewer pointed out, a statistically significant difference was observed at 90 min in mice with ablation of LC-NA neurons, but not in those with LC^→SDH^-NA neuron ablation. This is likely due to a slightly higher threshold in the control group at this time point (Figure 1K), and it remains unclear whether there is a mechanistic difference between the two groups at this specific time point.

(d) A valuable addition would be staining for noradrenergic terminals in the spinal cord for the intersectional approach (Figure 1J), as done in Figures 1F/G. LC projections terminate preferentially in the SDH, whereas A5 projections terminate in the deep dorsal horn (DDH). Staining could clarify whether circuits beyond the LC are being ablated.

As suggested, we performed DTR immunostaining in the SDH; however, we did not detect any DTR immunofluorescence there. A similar result was also observed in the spinal terminals of DTR-expressing primary afferent fibers (our unpublished data). The reason for this is unclear, but to the best of our knowledge, no studies have clearly shown DTR expression at presynaptic terminals, which may be because the action of DTX on the neuronal cell body is necessary for cell ablation. Nevertheless, as described in our response to the weakness (5)-f by Reviewer 1 (Public review), we have now confirmed the specific expression of DTR in the LC, but not in the A5 and A7 regions (Figure S4; lines 127-128).

(e) Furthermore, different LC neurons often mediate opposite physiological outcomes depending on their projection targets-for example, dorsal LC neurons projecting to the prefrontal cortex PFCx are pronociceptive, while ventral LC neurons projecting to the SC are antinociceptive (PMIDs: 29027903, 34344259, 36625030). Given this functional diversity, direct injection into the LC is likely to result in nonspecific effects.

To avoid behavioral outcomes resulting from a mixture of facilitatory and inhibitory effects caused by activating the entire population of LC-NA neurons, we employed a specific manipulation targeting LC^→SDH^-NA neurons using AAV vectors. The specificity of this manipulation was confirmed in our previous study (Kohro et al., Nat Neurosci, 2020 (PMID: 33020652)) and in the current study (Figure S4). Using this approach, we previously demonstrated that LC neurons can exert pronociceptive effects via astrocytes in the SDH (Kohro et al., Nat Neurosci, 2020 (PMID: 33020652)). This pronociceptive role is further supported by the current study, which uses a more selective manipulation of LC^→SDH^-NA neurons through a *NET-Cre* mouse line. In addition, intrathecal administration of relatively low doses of NA in naïve mice clearly induces mechanical pain hypersensitivity. Nevertheless, we have also acknowledged that several recent studies have reported an inhibitory role of LC^→SDH^-NA neurons in spinal nociceptive signaling. The reason for these differing behavioral outcomes remains unclear, but several methodological differences may underlie the discrepancy. First, the degree of LC^→SDH^-NA neuronal activity may play a role. Although direct comparisons between studies reporting pro- and anti-nociceptive effects are difficult, our previous studies demonstrated that intrathecal administration of high doses of NA in naïve mice does not induce mechanical pain hypersensitivity (Kohro et al., Nat Neurosci, 2020 (PMID: 33020652)). Second, the sensory modality used in behavioral testing may be a contributing factor as the pronociceptive effect of NA appears to be selectively observed in responses to mechanical, but not thermal, stimuli (Kohro et al., Nat Neurosci, 2020 (PMID: 33020652)). This sensory modality-selective effect is also evident in mice subjected to acute restraint stress (Table S1). Therefore, the role of LC^→SDH^-NA neurons in modulating nociceptive signaling in the SDH is more complex than previously appreciated, and their contribution to pain regulation should be reconsidered in light of factors such as NA levels, sensory modality, and experimental context. In revising the manuscript, we have included some points described above in the Discussion (lines 282-291).

Conclusion on Specificity: The authors are strongly encouraged to address these limitations directly, as they significantly affect the validity of the conclusions regarding the LC pathway. Providing more robust evidence, acknowledging experimental limitations, and incorporating complementary analyses would greatly strengthen the manuscript.

We appreciate the reviewer’s comments. We fully acknowledge the limitations raised and agree that addressing them directly is important for the rigor of our conclusions on the LC pathway. To this end, we have performed additional experiments (e.g., Figure A and S4), which are now included in the revised manuscript. Furthermore, we have also newly added a new paragraph for experimental limitations in the end of Discussion section (lines 373-408). We believe these new data substantially strengthen the validity of our findings and have clarified these points in the Discussion section.

(2) Discrepancies in Data(a) Figures 1B and 1E: The behavioural effect of stress on PWT (Figure 1E) persists for 120 minutes, whereas Ca2+ imaging changes (Figure 1B) are only observed in the first 20 minutes, with signal attenuation starting at 30 minutes. This discrepancy requires clarification, as it impacts the proposed mechanism.

Thank you for your important comment. As pointed out by the reviewer, there is a difference between the duration of behavioral responses and Ca^2+^ events, although the exact time point at which the PWT begins to decline remains undetermined (as behavioral testing cannot be conducted during stress exposure). A similar temporal difference was also observed following intraplantar injection of capsaicin (Kohro et al., Nat Neurosci, 2020 (PMID: 33020652)); while LC^→SDH^-NA neuron-mediated astrocytic Ca^2+^ responses in SDH astrocytes last for 5–10 min after injection, behavioral hypersensitivity peaks around 60 min post-injection and gradually returns to baseline over the subsequent 60–120 min. These findings raise the possibility that astrocyte-mediated pain hypersensitivity in the SDH may involve a sustained alteration in spinal neural function, such as central sensitization. We have added this hypothesis to the Discussion section of the revised manuscript (lines 399-408), as it represents an important direction for future investigation.

(b) Figure 4E: The effect is barely visible, and the tissue resembles "Swiss cheese," suggesting poor staining quality. This is insufficient for such an important conclusion. Improved staining and/or complementary staining (e.g., cFOS) are needed. Additionally, no clear difference is observed between Stress+Ab stim. and Stress+Ab stim.+CPT, raising doubts about the robustness of the data.

As suggested, we performed c-FOS immunostaining and obtained clearer results (Figure 4E,F; lines 243-252). We also quantitatively analyzed the number of c-FOS^+^ cells in the superficial laminae, and the results are consistent with those obtained from the pERK experiments.

(c) Discrepancy with Existing Evidence: The claim regarding the pronociceptive effect of LC→SDH-NAergic signalling on mechanical hypersensitivity contrasts with findings by Kucharczyk et al. (PMID: 35245374), who reported no facilitation of spinal convergent (wide-dynamic range) neuron responses to tactile mechanical stimuli, but potent inhibition to noxious mechanical von Frey stimulation. This discrepancy suggests alternative mechanisms may be at play and raises the question of why noxious stimuli were not tested.

In our experiments, ChrimsonR expression was observed in the superficial and deeper laminae of the spinal cord (Figure S6). Due to the technical limitations of the optical fibers used for optogenetics, the light stimulation could only reach the superficial laminae; therefore, it may not have affected the activity of neurons (including WDR neurons) located in the deeper laminae. Furthermore, the study by Kucharczyk et al. (Brain, 2022 (PMID: 35245374)) employed a stimulation protocol that differed from ours, applying continuous stimulation over several minutes. Given that the levels of NA released from LC^→SDH^-NAergic terminals in the SDH increase with the duration of terminal stimulation (as shown in Figure 2B), longer stimulation may result in higher levels of NA in the SDH. Considering also our data indicating that the pro- and anti-nociceptive effects of NA are dose dependent (Kohro et al., Nat Neurosci, 2020 (PMID: 33020652)), these differences may be related to LC^→SDH^-NA neuron activity, NA levels in the SDH, and the differential responses of SDH neurons in the superficial versus deeper laminae (lines 388-395).

(3) Sole reliance on Von Frey testingThe exclusive use of von Frey as a behavioural readout for mechanical sensitisation is a significant limitation. This assay is highly variable, and without additional supporting measures, the conclusions lack robustness. Incorporating other behavioural measures, such as the adhesive tape removal test to evaluate tactile discomfort, the needle floor walk corridor to assess sensitivity to uneven or noxious surfaces, or the kinetic weight-bearing test to measure changes in limb loading during movement, could provide complementary insights. Physiological tests, such as the Randall-Selitto test for noxious pressure thresholds or CatWalk gait analysis to evaluate changes in weight distribution and gait dynamics, would further strengthen the findings and allow for a more comprehensive assessment of mechanical sensitisation.

Thank you for your suggestion. Based on our previous findings that Hes5^+^ astrocytes in the SDH selectively modulate mechanosensory signaling (Kohro et al., Nat Neurosci, 2020 (PMID: 33020652)), the present study focused on behavioral responses to mechanical stimuli using von Frey filaments. As we have not previously conducted most of the behavioral tests suggested by the reviewers, and as we currently lack the necessary equipments for these tests (e.g., Randall–Selitto test, CatWalk gait analysis, and weight-bearing test), we were unable to include them in this study. However, it will be of great interest in future research to investigate whether activation of the LC^→SDH^-NA neuron-to-SDH Hes5^+^ astrocyte signaling pathway similarly sensitizes behavioral responses to other types of mechanical stimuli and also to investigate the sensory modality-selective pro- and antinociceptive role of LC^→SDH^-NAergic signaling in the SDH (lines 396-399).

Overall ConclusionThis study addresses an important and complex topic with innovative methods and compelling data. However, the conclusions rely on several assumptions that require more robust evidence. Specificity of the LC pathway, experimental discrepancies, and methodological limitations (e.g., sole reliance on von Frey) must be addressed to substantiate the claims. With these issues resolved, this work could significantly advance our understanding of astrocytic and noradrenergic contributions to pain modulation.

We have made every effort to address the reviewer’s concerns through additional experiments and analyses. Based on the new control data presented, we believe that our explanation is reasonable and acceptable. Although additional data cannot be provided on some points due to methodological constraints and limitations of the techniques currently available in our laboratory, we respectfully submit that the evidence presented sufficiently supports our conclusions.

**Reviewer #3 (Recommendations for the authors):**
A lot of beautiful and challenging-to-collect data is presented. Sincere congratulations to all the authors on this achievement!Notwithstanding, please carefully reconsider the conclusions regarding the LC pathway, as additional evidence is required to ensure their specificity and robustness.

We thank the reviewer for the kind comments and for raising an important point regarding the LC pathway. The reviewer’s feedback prompted us to conduct additional investigations to further strengthen the validity of our conclusions. We have incorporated these new data and analyses into the revised manuscript, and we believe that these revisions substantially enhance the robustness and reliability of our findings.